# Sequencer: Deep LSTM for Image Classification

**Yuki Tatsunami**[1,2]   **Masato Taki**[1]
[1]Rikkyo University, Tokyo, Japan
[2]AnyTech Co., Ltd., Tokyo, Japan
{y.tatsunami, taki_m}@rikkyo.ac.jp

## Abstract

In recent computer vision research, the advent of the Vision Transformer (ViT) has rapidly revolutionized various architectural design efforts: ViT achieved state-of-the-art image classification performance using self-attention found in natural language processing, and MLP-Mixer achieved competitive performance using simple multi-layer perceptrons. In contrast, several studies have also suggested that carefully redesigned convolutional neural networks (CNNs) can achieve advanced performance comparable to ViT without resorting to these new ideas. Against this background, there is growing interest in what inductive bias is suitable for computer vision. Here we propose *Sequencer*, a novel and competitive architecture alternative to ViT that provides a new perspective on these issues. Unlike ViTs, Sequencer models long-range dependencies using LSTMs rather than self-attention layers. We also propose a two-dimensional version of Sequencer module, where an LSTM is decomposed into vertical and horizontal LSTMs to enhance performance. Despite its simplicity, several experiments demonstrate that Sequencer performs impressively well: Sequencer2D-L, with 54M parameters, realizes 84.6% top-1 accuracy on only ImageNet-1K. Not only that, we show that it has good transfer-ability and the robust resolution adaptability on double resolution-band. Our source code is available at https://github.com/okojoalg/sequencer.

## 1 Introduction

The de-facto standard for computer vision has been convolutional neural networks (CNNs) [39, 64, 22, 65, 66, 9, 29, 67]. However, inspired by the many breakthroughs in natural language processing (NLP) achieved by Transformers [75, 35, 57], applications of Transformers for computer vision are now being actively studied. In particular, Vision Transformer (ViT) [16] is a pure Transformer applied to image recognition and achieves performance competitive with CNNs. Various studies triggered by ViT have shown that the state-of-the-art (SOTA) performance can be achieved for a wide range of vision tasks using self-attention alone [79, 48, 73, 47, 15], without convolution.

The reason for this success is thought to be due to the ability of self-attention to model long-range dependencies. However, it is still unclear how essential the

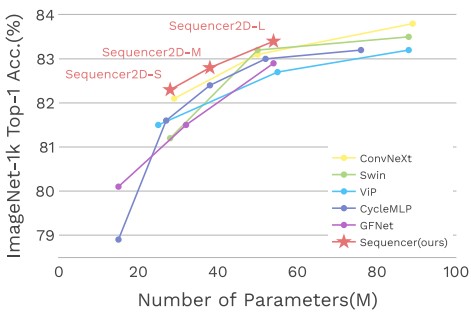

Figure 1: **IN-1K top-1 accuracy v.s. model parameters.** All models are trained on IN-1K at resolution $224^2$ from scratch.

self-attention is to the effectiveness of Transformers for vision tasks. Indeed, the MLP-Mixer [70] based only on multi-layer perceptrons (MLPs) is proposed as an appealing alternative to Vision Trans-

36th Conference on Neural Information Processing Systems (NeurIPS 2022).

formers (ViTs). In addition, some studies [49, 14] have shown that carefully designed CNNs are still competitive enough with Transformers in computer vision. Therefore, identifying which architectural designs are inherently effective for computer vision tasks is of great interest for current research [83]. This paper provides a new perspective on this issue by proposing a novel and competitive alternative to these vision architectures.

We propose the *Sequencer* architecture, that uses the long short-term memory (LSTM) [27] rather than the self-attention for sequence modeling. The macro-architecture design of Sequencer follows ViTs, which iteratively applies token mixing and channel mixing, but the self-attention layer is replaced by one based on LSTMs. In particular, Sequencer uses bidirectional LSTM (BiLSTM) [63] as a building block. While simple BiLSTM shows a certain level of performance, Sequencer can be further improved by using ideas similar to Vision Permutator (ViP) [28]. The key idea in ViP is to process the vertical and horizontal axes in parallel. We also introduce two BiLSTMs for top/bottom and left/right directions in parallel. This modification improves the efficiency and accuracy of Sequencer because this structure reduces the length of the sequence and yields a spatially meaningful receptive field.

When pre-trained on ImageNet-1K (IN-1K) dataset, our new attention-free architecture outperforms advanced architectures such as Swin [48] and ConvNeXt [49] of comparable size, see Figure 1. It also outperforms other attention-free and CNN-free architectures such as MLP-Mixer [70] and GFNet [61], making Sequencer an attractive new alternative to the self-attention mechanism in vision tasks.

This study also aims to propose novel architecture with practicality by employing LSTM for spatial pattern processing. Notably, Sequencer exhibits robust resolution adaptability, which strongly prevents accuracy degradation even when the input's resolution is increased double during inference. Moreover, fine-tuning Sequencer on high-resolution data can achieve higher accuracy than Swin-B [48] and Sequencer is also useful for semantic segmentation. On peak memory, Sequencer tends to be more economical than ViTs and recent CNNs for high-resolution input. Although Sequencer requires more FLOPs than other models due to recursion, the higher resolution improves the relative efficiency of peak memory, enhancing the accuracy/cost trade-off at a high-resolution regime. Therefore, Sequencer also has attractive properties as a practical image recognition model.

## 2 Related works

Inspired by the success of Transformers in NLP [75, 35, 57, 58, 3, 60], various applications of self-attention have been studied in computer vision. For example, in iGPT [6], an attempt was made to apply autoregressive pre-training with causal self-attention [57] to image classification. However, due to the computational cost of pixel-wise attention, it could only be applied to low-resolution images, and its ImageNet classification performance was significantly inferior to the SOTA. ViT [16], on the other hand, quickly brought Transformer's image classification performance closer to SOTA with its idea of applying bidirectional self-attention [35] to image patches rather than pixels. Various architectural and training improvements [72, 84, 79, 90, 48, 73, 5] have been attempted for ViT [16]. In this paper, we do not improve self-attention itself but propose a completely new module for image classification to replace it.

The extent to which attention-based cross-token communication inherently contributes to ViT's success is not yet well understood, starting with MLP-Mixer [70], which completely replaced ViT's self-attention with MLP, various MLP-based architectures [71, 46, 28, 69, 68, 13] have achieved competitive performance on the ImageNet dataset. We refer to these architectures as global MLPs (GMLPs) because they have global receptive fields. This series of studies cast doubt on the need for self-attention. From a practical standpoint, however, these MLP-based models have a drawback: they need to be finetuned to cope with flexible input sizes during inference by modifying the shape of their token-mixing MLP blocks. This resolution adaptability problem has been improved in CycleMLP [7], for example, by the idea of realizing a local kernel with a cyclic MLP. There are similar ideas such as [82, 81, 42, 21] which are collectively referred to as local MLPs (LMLPs). Besides the MLP-based idea, several other interesting self-attention alternatives have been found. GFNet [61] uses Fourier transformation of the tokens and mixes the tokens by global filtering in the frequency domain. PoolFormer [83], on the other hand, achieved competitive performance with only local pooling of tokens, demonstrating that simple local operations are also a suitable alternative. Our proposed Sequencer is a new alternative to self-attention that differs from both of the above, and

Sequencer is an attempt to realize token mixing in vision architectures using only LSTM. It achieved competitive performance with SOTA on the IN-1K benchmark, especially with an architecture that can flexibly adapt to higher resolution.

The idea of spatial axis decomposition has been used several times in neural architecture in computer vision. For example, SqueezeNeXt [17] decomposes a 3x3 convolution layer into 1x3 and 3x1 convolution layers, resulting in a lightweight model. Criss-cross attention [31] reduces memory usage and computational complexity by restricting the attention to only vertical and horizontal portions. Current architectures such as CSwin [15], Couplformer [40], ViP [28], RaftMLP [69], SparseMLP [68], and MorphMLP [86] have included similar ideas to improve efficiency and performance.

In the early days of deep learning, there were attempts to use RNNs for image recognition. The earliest study that applied RNNs to image recognition is [19]. The primary difference between our study and [19] is that we utilize a usual RNN in place of a 2-multi-dimensional RNN(2MDRNN). The 2MDRNN requires $H + W$ sequential operations; The LSTM requires $H$ sequential operations, where $H$ and $W$ are height and width, respectively. For subsequent work on image recognition using 2MDRNNs, see [20, 32, 4, 43]. [4] proposed an architecture in which information is collected from four directions (upper left, lower left, upper right, and lower right) by RNNs for understanding natural scene images. [43] proposed a novel 2MDRNN for semantic object parsing that integrates global and local context information, called LG-LSTM. The overall architecture design is structured to input deep ConvNet features into the LG-LSTM, unlike Sequencer which stacks LSTMs. ReNet [77] is most relevant to our work; ReNet [77] uses a 4-way LSTM and non-overlapping patches as input. In this respect, it is similar to Sequencer. Meanwhile, there are three differences. First, Sequencer is the first MetaFormer [83] realized by adopting LSTM as the token mixing block. Sequencer also adopts a larger patch size than ReNet [77]. The benefit of adopting these designs is that we can modernize LSTM-based vision architectures and fairly compare LSTM-based models with ViT. As a result, our results provide further evidence for the extremely interesting hypothesis MetaFormer [83]. Second, the way vertical BiLSTMs and horizontal BiLSTMs are connected is different. Our work connects them in parallel, allowing us to gather vertical and horizontal information simultaneously. On the other hand, in ReNet [77], the output of the horizontal BiLSTM is used as input to the vertical BiLSTM. Finally, we trained Sequencer on large datasets such as ImageNet, whereas ReNet [77] is limited to small datasets as MNIST [41], CIFAR-10 [38], and SVHN [54], and has not shown the effectiveness of LSTM for larger datasets. ReSeg [76] applied ReNet to semantic segmentation. RNNs have been applied not only to image recognition, but also to generative models: PixelRNN [74] is a pixel-channel autoregressive generative model of images using Row RNN, which consists of a 1D-convolution and a usual RNN, and Diagonal BiLSTM, which is computationally expensive.

In NLP, attempts have been made to avoid the computational cost of attention by approximating causal self-attention with recurrent neural network (RNN) [34] or replacing it with RNN after training [33]. In particular, in [34], an autoregressive pixel-wise image generation task is experimented with an architecture where the attentions in iGPT are approximated by RNNs. These studies are specific to unidirectional Transformers, in contrast to our token-based Sequencer which is the bidirectional analog of them.

## 3 Method

In this section, we briefly recap the preliminary background on LSTM and further describe the details of the proposed architectures.

### 3.1 Preliminaries: Long short-term memory

LSTM [27] is a specialized recurrent neural network (RNN) for modeling long-term dependencies of sequences. Plain LSTM has an input gate $i_t$ that controls the storage of inputs, a forget gate $f_t$ that controls the forgetting of the former cell state $c_{t-1}$ and an output gate $o_t$ that controls the cell output $h_t$ from the current cell state $c_t$. Plain LSTM is formulated as follows:

$$i_t = \sigma\left(W_{xi}x_t + W_{hi}h_{t-1} + b_i\right), \quad f_t = \sigma\left(W_{xf}x_t + W_{hf}h_{t-1} + b_f\right), \tag{1}$$

$$c_t = f_t \odot c_{t-1} + i_t \odot \tanh\left(W_{xc}x_t + W_{hc}h_{t-1} + b_c\right), \quad o_t = \sigma\left(W_{xo}x_t + W_{ho}h_{t-1} + b_o\right), \tag{2}$$

$$h_t = o_t \odot \tanh(c_t), \tag{3}$$

where $\sigma$ is the logistic sigmoid function and $\odot$ is Hadamard product.

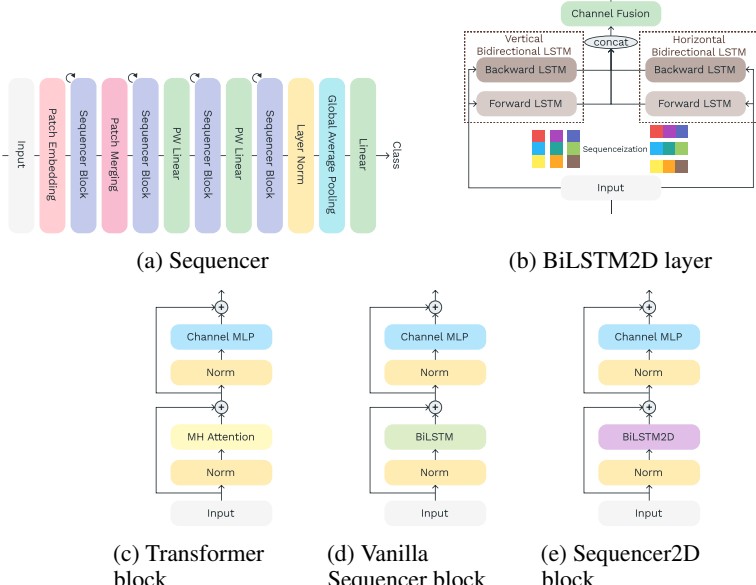

(a) Sequencer

(b) BiLSTM2D layer

(c) Transformer block

(d) Vanilla Sequencer block

(e) Sequencer2D block

Figure 2: (a) The architecture of Sequencers; (b) The figure outlines the BiLSTM2D layer, which is the main component of Sequencer2D. (c) Transformer block consist of multi-head attention. In contrast, (d) Vanilla Sequencer block and (e) Sequencer2D block, utilized on our archtecture, composed of BiLSTM or BiLSTM2D instead of multi-head attention.

BiLSTM [63] is profitable for sequences where mutual dependencies are expected. A BiLSTM consists of two plain LSTMs. Let $\overrightarrow{x}$ be the input series and $\overleftarrow{x}$ be the rearrangement of $\overrightarrow{x}$ in reverse order. $\overrightarrow{h}_{\text{for}}$ and $\overleftarrow{h}_{\text{back}}$ are the outputs obtained by processing $\overrightarrow{x}$ and $\overleftarrow{x}$ with the corresponding LSTMs, respectively. Let $\overrightarrow{h}_{\text{back}}$ be the output $\overleftarrow{h}_{\text{back}}$ rearranged in the original order, and the output of BiLSTM is obtained as follows:

$$\overrightarrow{h}_{\text{for}}, \overleftarrow{h}_{\text{back}} = \text{LSTM}_{\text{for}}(\overrightarrow{x}), \text{LSTM}_{\text{back}}(\overleftarrow{x}), \quad h = \texttt{concatenate}(\overrightarrow{h}_{\text{for}}, \overrightarrow{h}_{\text{back}}). \quad (4)$$

Assume that both $\overrightarrow{h}_{\text{for}}$ and $\overrightarrow{h}_{\text{back}}$ have the same hidden dimension $D$, which is hyperparameter of BiLSTM. Accordingly, vector $h$ has dimension $2D$.

## 3.2 Sequencer architecture

**Overall architecture** In the last few years, ViT and its many variants based on self-attention [16, 72, 48, 91] have attracted much attention in computer vision. Following these, several works [70, 71, 46, 28] have been proposed to replace self-attention with MLP. There have also been studies of replacing self-attention with a hard local induced bias module [7, 83] and with a global filter [61] using the fast Fourier transform algorithm (FFT) [10]. This paper continues this trend and attempts to replace the self-attention layer with LSTM [27]: we propose a new architecture aiming at memory saving by mixing spatial information with LSTM, which is memory-economical compared to ViT, parameter-saving, and has the ability to learn long-range dependencies.

Figure 2a shows the overall structure of Sequencer architecture. Sequencer architecture takes non-overlapping patches as input and projects them onto the feature map. Sequencer block, which is a core component of Sequencer, consists of the following sub-components: (1) BiLSTM layer can mix spatial information more memory-economically for high-resolution images than Transformer layer and more globally than CNN. (2) MLP for channel-mixing as well as [16, 70]. Sequencer block is called Vanilla Sequencer block when plain BiLSTM layers are used as BiLSTM layers as Figure 2d and Sequencer2D block when BiLSTM2D layers are used as Figure 2e. We define BiLSTM2D layer later. The output of the last block is sent to the linear classifier via the global average pooling layer, as in most other architectures.

**BiLSTM2D layer** We propose the BiLSTM2D layer as a technique to mix 2D spatial information efficaciously. It has two plain BiLSTMs: a vertical BiLSTM and a horizontal one. For an input $\mathbf{X} \in \mathbb{R}^{H \times W \times C}$, $\{\mathbf{X}_{:,w,:} \in \mathbb{R}^{H \times C}\}_{w=1}^{W}$ is viewed as a set of sequences, where $H$ is the number of tokens in the vertical direction, $W$ is the number of sequences in the horizontal direction, and $C$ is the channel dimension. All sequences $\mathbf{X}_{:,w,:}$ are input into the vertical BiLSTM with shared weights and hidden dimension $D$:

$$\mathbf{H}_{:,w,:}^{\text{ver}} = \text{BiLSTM}(\mathbf{X}_{:,w,:}). \tag{5}$$

In a very similar manner, $\{\mathbf{X}_{h,:,:} \in \mathbb{R}^{W \times C}\}_{h=1}^{H}$ is viewed as a set of sequences, and all sequences $\mathbf{X}_{h,:,:}$ are input into the horizontal BiLSTM with shared weights and hidden dimension $D$ as well:

$$\mathbf{H}_{h,:,:}^{\text{hor}} = \text{BiLSTM}(\mathbf{X}_{h,:,:}). \tag{6}$$

We combine $\{\mathbf{H}_{:,w,:}^{\text{ver}} \in \mathbb{R}^{H \times 2D}\}_{w=1}^{W}$ into $\mathbf{H}^{\text{ver}} \in \mathbb{R}^{W \times H \times 2D}$ and $\{\mathbf{H}_{h,:,:}^{\text{hor}} \in \mathbb{R}^{W \times 2D}\}_{h=1}^{H}$ into $\mathbf{H}^{\text{hor}} \in \mathbb{R}^{W \times H \times 2D}$. They are then concatenated and processed point-wisely in a fully-connection layer. These processes are formulated as follows:

$$\mathbf{H} = \text{concatenate}(\mathbf{H}^{\text{ver}}, \mathbf{H}^{\text{hor}}), \quad \hat{\mathbf{X}} = \text{FC}(\mathbf{H}), \tag{7}$$

where $\text{FC}(\cdot)$ denotes the fully-connected layer with weight $W \in \mathbb{R}^{C \times 4D}$. The `PyTorch`-like pseudocode is shown in Appendix B.1.

BiLSTM2D is more memory-economical and throughput-efficiency than multi-head-attention of ViT for high-resolution input. BiLSTM2D involves $(WC + HC)/2$ dimensional cell states, while a multi-head-attention involves $h * (HW)^2$ dimensional attention map where $h$ is a number of heads. Thus, as $H$ and $W$ increase, the memory cost of an attention map increases more rapidly than the cost of a cell state. On throughput, the computational complexity of self-attention is $\mathcal{O}(W^4 C)$, whereas the computational complexity of BiLSTM is $\mathcal{O}(WC^2)$ where we assume $W = H$ for simplicity. There are $\mathcal{O}(W)$ sequential operations for BiLSTM2D. Therefore, assuming we use a sufficiently efficient LSTM cell implementation, such as official PyTorch LSTMs we are using, the increase of the complexity of self-attention is much more rapid than BiLSTM2D. It implies a lower throughput of attention compared to BiLSTM2D. See an experiment in Section 4.5.

**Architecture variants** For comparison between models of different depths consisting of Sequencer2D blocks, we have prepared three models with different depths: 18, 24, and 36. The names of the models are *Sequencer2D-S*, *Sequencer2D-M*, and *Sequencer2D-L*, respectively. The hidden dimension is set to $D = C/4$. Details of these models are provided in Appendix B.2.

As shown in subsection 4.1, these architectures outperform typical models. Interestingly, however, subsection 4.3 shows that replacing Sequencer2D block with the simpler Vanilla Sequencer block maintains moderate accuracy. We denote such a model as Vanilla Sequencer. Note that some of the explicit positional information is lost in the *Vanilla Sequencer* because the model treats patches as a 1D sequence.

## 4 Experiments

In this section, we compare Sequencers with previous studies on the IN-1K benchmark [39]. We also carry out ablation studies, transfer learning studies, and analysis of the results to demonstrate the effectiveness of Sequencers. We adopt `PyTorch` [56] and `timm` [80] library to implement models in the conduct of all experiments. See Appendix B for more setup details.

### 4.1 Scratch training on IN-1K

We utilize IN-1K [39], which has 1000 classes and contains 1,281,167 training images and 50,000 validation images. We adopt AdamW optimizer [50]. Following the previous study [72], we adopt the base learning rate $\frac{\text{batch size}}{512} \times 5 \times 10^{-4}$. The batch sizes for Sequencer2D-S, Sequencer2D-M, and Sequencer2D-L are 2048, 1536, and 1024, respectively. As a regularization method, stochastic depth [30] and label smoothing [66] are employed. As data augmentation methods, mixup [87], cutout [12], cutmix [85], random erasing [88], and randaugment [11] are applied.

Table 1: The table shows the top-1 accuracy when trained on IN-1K, comparing our model with other similar scale representative models. Training and inference throughput and their peak memory were measured with 16 images per batch on a single V100 GPU. The left sides of the slashes are values during training, and the right sides of the slashes are values during inference. Fine-tuned models marked with "↑". Note Sequencer2D-L↑ are compared to Swin-B↑ and ConvNeXt-B↑ with more parameters since Swin and ConvNeXt have not fine-tuned models of similar parameters with Sequencer2D-L↑ in the original papers.

| Model | Family | Res. | #Param. | FLOPs | Throughput (image/s) | Peak Mem. (MB) | Top-1 Acc.(%) | Pre FT Top-1 Acc.(%) |
|---|---|---|---|---|---|---|---|---|
| Training from scratch | | | | | | | | |
| RegNetY-4GF [59] | CNN | $224^2$ | 21M | 4.0G | 228/823 | 1136/225 | 80.0 | non-fine-tune |
| ConvNeXt-T [49] | CNN | $224^2$ | 29M | 4.5G | 337/1124 | 1418/248 | 82.1 | as above |
| DeiT-S [72] | Trans. | $224^2$ | 22M | 4.6G | 480/1569 | 1195/180 | 79.9 | as above |
| Swin-T [48] | Trans. | $224^2$ | 28M | 4.5G | 268/894 | 1613/308 | 81.2 | as above |
| ViP-S/7 [28] | GMLP | $224^2$ | 25M | 6.9G | 214/702 | 1587/195 | 81.5 | as above |
| CycleMLP-B2 [7] | LMLP | $224^2$ | 27M | 3.9G | 158/586 | 1357/234 | 81.6 | as above |
| PoolFormer-S24 [83] | LMLP | $224^2$ | 21M | 3.6G | 313/988 | 1461/183 | 80.3 | as above |
| Sequencer2D-S | Seq. | $224^2$ | 28M | 8.4G | 110/347 | 1799/196 | **82.3** | as above |
| RegNetY-8GF [59] | CNN | $224^2$ | 39M | 8.0G | 211/751 | 1776/333 | 81.7 | as above |
| T2T-ViT$_t$-19 [84] | Trans. | $224^2$ | 39M | 9.8G | 197/654 | 3520/1140 | 82.2 | as above |
| CycleMLP-B3 [7] | LMLP | $224^2$ | 38M | 6.9G | 100/367 | 2326/287 | 82.6 | as above |
| PoolFormer-S36 [83] | LMLP | $224^2$ | 31M | 5.2G | 213/673 | 2187/220 | 81.4 | as above |
| GFNet-H-S [61] | FFT | $224^2$ | 32M | 4.5G | 227/755 | 1740/282 | 81.5 | as above |
| Sequencer2D-M | Seq. | $224^2$ | 38M | 11.1G | 83/270 | 2311/244 | **82.8** | as above |
| RegNetY-12GF [59] | CNN | $224^2$ | 46M | 12.0G | 199/695 | 2181/440 | 82.4 | as above |
| ConvNeXt-S [49] | CNN | $224^2$ | 50M | 8.7G | 212/717 | 2265/341 | 83.1 | as above |
| Swin-S [48] | Trans. | $224^2$ | 50M | 8.7G | 165/566 | 2635/390 | 83.2 | as above |
| Mixer-B/16 [70] | GMLP | $224^2$ | 59M | 12.7G | 338/1011 | 1864/407 | 76.4 | as above |
| ViP-M/7 [28] | GMLP | $224^2$ | 55M | 16.3G | 130/395 | 3095/396 | 82.7 | as above |
| CycleMLP-B4 [7] | LMLP | $224^2$ | 52M | 10.1G | 70/259 | 3272/338 | 83.0 | as above |
| PoolFormer-M36 [83] | LMLP | $224^2$ | 56M | 9.1G | 171/496 | 3191/368 | 82.1 | as above |
| GFNet-H-B [61] | FFT | $224^2$ | 54M | 8.4G | 144/482 | 2776/367 | 82.9 | as above |
| Sequencer2D-L | Seq. | $224^2$ | 54M | 16.6G | 54/173 | 3516/322 | **83.4** | as above |
| Fine-tuning | | | | | | | | |
| ConvNeXt-B↑ [49] | CNN | $384^2$ | 89M | 45.1G | 78/234 | 7329/870 | 85.1(+1.3) | 83.8 |
| Swin-B↑ [48] | Trans. | $384^2$ | 88M | 47.1G | 54/156 | 12933/1532 | 84.5(+1.0) | 83.5 |
| GFNet-B↑ [61] | FFT | $384^2$ | 47M | 23.2G | 137/390 | 3710/416 | 82.1(+0.8) | 82.9 |
| Sequencer2D-L↑ | Seq. | $392^2$ | 54M | 50.7G | 26/84 | 9062/481 | 84.6(+1.2) | 83.4 |

Table 1 shows the results that are comparing the proposed models to others with a comparable number of parameters to our models, including models with local and global receptive fields such as CNNs, ViTs, and MLP-based and FFT-based models. Sequencers has the disadvantage that its throughput is slower than other models because it uses RNNs. In the scratch training on IN-1K, however, they outperform these recent comparative models in accuracy across their parameter bands. In particular, Seqeuncer2D-L is competitive with recently discussed models with comparable parameters such as ConvNeXt-S [49] and Swin-S [48], with accuracy outperformance of 0.3% and 0.2%, respectively.

Table 1 demonstrates that Sequencer's throughput is not good. The training throughput is about three times the inference throughput for all these models. Compared to other models, both measured inference and training time are not good.

## 4.2 Fine-tuning on IN-1K

In this fine-tuning study, Sequencer2D-L pre-trained on IN-1K at $224^2$ resolution is fine-tuned on IN-1K at $392^2$ resolution. We compare it with the other models fine-tuned on IN-1K at $384^2$

Table 2: **Sequencer ablation experiments**. We adopt Sequencer2D-S variant for these ablation studies. **C1** denotes vertical BiLSTM, **C2** denotes horizontal BiLSTM, and **C3** denotes channel fusion component. When vertical BiLSTM only, horizontal BiLSTM only or unidirectional BiLSTM2D, its hidden dimension needs to be doubled from the original setting because it compensates the output dimension for the excluded LSTM and matches the dimensions.

<table>
<tr><td colspan="4" align="center">(a) Components</td><td colspan="2" align="center">(b) LSTM Direction</td><td colspan="4" align="center">(c) Vanilla Sequencer</td></tr>
<tr><td>C1</td><td>C2</td><td>C3</td><td>Acc.</td><td>Bidirectional</td><td>Acc.</td><td>Model</td><td>#Params.</td><td>FLOPs</td><td>Acc.</td></tr>
<tr><td>✓</td><td></td><td></td><td>75.6</td><td></td><td>79.7</td><td>VSequencer-S</td><td>33M</td><td>8.4G</td><td>78.0</td></tr>
<tr><td></td><td>✓</td><td></td><td>75.0</td><td>✓</td><td>**82.3**</td><td>VSequencer(H)-S</td><td>28M</td><td>8.4G</td><td>78.8</td></tr>
<tr><td>✓</td><td>✓</td><td></td><td>81.6</td><td></td><td></td><td>VSequencer(PE)-S</td><td>33M</td><td>8.4G</td><td>78.1</td></tr>
<tr><td>✓</td><td>✓</td><td>✓</td><td>**82.3**</td><td></td><td></td><td>Sequencer2D-S</td><td>28M</td><td>8.4G</td><td>**82.3**</td></tr>
</table>

<table>
<tr><td colspan="4" align="center">(d) Hidden dimension</td><td colspan="4" align="center">(e) Various RNNs</td></tr>
<tr><td>Hidden dim. ratio</td><td>#Params.</td><td>FLOPs</td><td>Acc.</td><td>Model</td><td>#Params.</td><td>FLOPs</td><td>Acc.</td></tr>
<tr><td>1x</td><td>**28M**</td><td>**8.4G**</td><td>82.3</td><td>RNN-Sequencer2D</td><td>19M</td><td>5.8G</td><td>80.6</td></tr>
<tr><td>2x</td><td>45M</td><td>13.9G</td><td>**82.6**</td><td>GRU-Sequencer2D</td><td>25M</td><td>7.5G</td><td>**82.3**</td></tr>
<tr><td></td><td></td><td></td><td></td><td>Seqeucer2D-S</td><td>28M</td><td>8.4G</td><td>**82.3**</td></tr>
</table>

resolution. Since 392 is divisible by 14, the input at this resolution can be split into patches without padding. However, note that this is not the case with a resolution of $384^2$.

As Table 1 indicates, even when higher-resolution Sequencer is fine-tuned, it is competitive with the latest models such as ConvNeXt [49], Swin [48], and GFNet [61].

## 4.3 Ablation studies

This subsection presents ablation studies based on Sequencer2D-S for further understanding of Sequencer. We seek to clarify the effectiveness and validity of the Sequencers architecture in terms of the importance of each component, bidirectional necessaries, setting of the hidden dimension, and the comparison with simple BiLSTM.

We show where and how relevant the components of BiLSTM2D are: The BiLSTM2D is composed of vertical BiLSTM, horizontal BiLSTM, and channel fusion elements. We want to see the validity of vertical BiLSTM, horizontal BiLSTM, and channel fusion. For this purpose, we examine the removal of channel fusion and vertical or horizontal BiLSTM. Table 2a shows the results. Removing channel fusion shows that the performance degrades from 82.3% to 81.6%. Furthermore, the additional removal of vertical or horizontal BiLSTM exposes a 6.0% or 6.6% performance drop, respectively. Hence, each component discussed here is necessary for Sequencer2D.

We show that the bidirectionality for BiLSTM2D is important for Sequencer. We compare Sequencer2D-S with a version that replaces the vertical and horizontal BiLSTMs with vertical and horizontal unidirectional LSTMs. Table 2b shows that the unidirectional model is 2.6% less accurate than the bidirectional model. This result attests to the significance of using not unidirectional LSTM but BiLSTM.

It is important to set the hidden dimension of LSTM to a reasonable size. As described in subsection 3.2, Sequencer2D sets the hidden dimension $D$ of BiLSTM to $D = C/4$, but this is not necessary if the model has channel fusions. Table 2d compares Sequencer2D-S with the model with increased $D$. Although accuracy is 0.3% improved, FLOPs increase by 65%, and the number of parameters increases by 60%. Namely, the accuracy has not improved for the increase in FLOPs. Moreover, the increase in dimension causes overfitting, which is discussed in Appendix C.3.

Vanilla Sequencer can also achieve accuracy that outperforms MLP-Mixer [70], but is not as accurate as Sequencer2D. Following experimental result supports the claim. We experiment with the Sequencer2D-S variants, where Vanilla Sequencer blocks replace the Sequencer2D blocks, called VSequencer-S(H), with incomplete positional information. In addition, we experiment with a variant of VSequencer-S(H) without the hierarchical structure, which we call VSequencer-S. VSequencer-S(PE) is VSequencer-S using ViTs-style learned positional embedding (PE) [16]. Table 2c indicates effectiveness for combination of LSTM and ViTs-like architecture. Surprisingly, even with Vanilla

Table 3: **Left.** Results on transfer learning. We transfer models trained on IN-1K to datasets from different domains. Sequencers use $224^2$ resolution images, while ViT-B/16 and EfficientNet-B7 work on higher resolution, see Res. column. **Right.** Semantic segmentation results on ADE20K [89]. All models are Semantic FPN [36] based. We show mIoU for the ADE20k validation set.

| Model | Res. | #Pr. | FLOPs | $CF_{10}$ | $CF_{100}$ | Flowers | Cars |
|---|---|---|---|---|---|---|---|
| ResNet50 [22] | $224^2$ | 26M | 4.1G | - | - | 96.2 | 90.0 |
| EN-B7 [67] | $600^2$ | 26M | 37.0G | 98.9 | 91.7 | 98.8 | 94.7 |
| ViT-B/16 [16] | $384^2$ | 86M | 55.4G | 98.1 | 87.1 | 89.5 | - |
| DeiT-B [72] | $224^2$ | 86M | 17.5G | 99.1 | 90.8 | 98.4 | 92.1 |
| CaiT-S-36 [73] | $224^2$ | 68M | 13.9G | 99.2 | 92.2 | 98.8 | 93.5 |
| ResMLP-24 [71] | $224^2$ | 30M | 6.0G | 98.7 | 89.5 | 97.9 | 89.5 |
| GFNet-H-B [61] | $224^2$ | 54M | 8.6G | 99.0 | 90.3 | 98.8 | 93.2 |
| Sequencer2D-S | $224^2$ | 28M | 8.4G | 99.0 | 90.6 | 98.2 | 93.1 |
| Sequencer2D-M | $224^2$ | 38M | 11.1G | 99.1 | 90.8 | 98.2 | 93.3 |
| Sequencer2D-L | $224^2$ | 54M | 16.6G | 99.1 | 91.2 | 98.6 | 93.1 |

| Model | #Pr. | mIoU |
|---|---|---|
| PVT-Small [79] | 28M | 39.8 |
| PoolFormer-S24 [83] | 23M | 40.3 |
| Sequencer2D-S | 32M | 46.1 |
| PVT-Medium [79] | 48M | 41.6 |
| PoolFormer-S36[83] | 35M | 42.0 |
| Sequencer2D-M | 42M | 47.3 |
| PVT-Large [79] | 65M | 42.1 |
| PoolFormer-M36 [83] | 60M | 42.4 |
| Sequencer2D-L | 58M | 48.6 |

Sequencer and Vanilla Sequencer(H) without PE, the performance reduction from Sequencer2D-S is only 4.3% and 3.5%, respectively. According to these results, there is no doubt that Vanilla Sequencer using BiLSTMs is significant enough, although not as accurate as Sequencer2D.

All LSTMs in the BiLSTM2D layer can be replaced with other recurrent networks such as gated recurrent units (GRUs) [8] or tanh-RNNs to define BiGRU2D layer or BiRNN2D layer. We also trained these models on IN-1K, so see Table 2e for the results. The table suggests that all of these variants, including RNN-cell, work well. Also, tanh-RNN performs slightly worse than others, probably due to its lower ability to model long-range dependence.

## 4.4 Transfer learning and semantic segmentation

Sequencers perform well on IN-1K, and they have good transferability. In other words, they have satisfactory generalization performance for a new domain, which is shown below. We utilize the commonly used CIFAR-10 [38], CIFAR-100 [38], Flowers-102 [55], and Stanford Cars [37] for this experiment. See the references and Appendix B.4 for details on the datasets. The results of the proposed model and the results in previous studies of models with comparable capacity are presented in Table 3. In particular, Sequencer2D-L achieves results that are competitive with CaiT-S-36 [73] and EfficientNet-B7 [67].

We experiment for semantic segmentation on ADE20K[89] dataset. See Appendix C.4 for details on the setup. Sequencer outperforms PVT [79] and PoolFormer [83] with similar parameters; compared to PoolFormer, mIoU is about 6 pts higher.

We have investigated a commonly object detection model with Sequencer as the backbone. Its performance is not much different from the case of ResNet [22] backbone. Its improvement is the future work. See Appendix C.5.

## 4.5 Analysis and visualization

In this subsection, we investigate the properties of Sequencer in terms of resolution adaptability and efficiency. Furthermore, effective receptive field (ERF) [51] and visualization of the hidden states provides insight into the question of how Sequencer recognizes images.

One of the attractive properties of Sequencer is its flexible adaptability to the resolution, with minimal impact on accuracy even when the resolution of the input image is varied from one-half to twice. In comparison, architectures like MLP-Mixer [70] have a fixed input resolution, and GFNet [61] requires interpolation of weights in the Fourier domain when inputting images with a resolution different from training images. We evaluate the resolution adaptability of models comparatively by inputting different resolution images to each model, without fine-tuning, with pre-trained weights on IN-1K at the resolution of $224^2$. Figure 3a compares absolute top-1 accuracy on IN-1K, and Figure 3b compares relative one to the input image with the resolution of $224^2$. By increasing the resolution by 28 for Sequencer2D-S and by 32 for other models, we avoid padding and prevent

the effect of padding on accuracy. Compared to DeiT-S [72], GFNet-S [61], CycleMLP-B2 [7], and ConvNeXt-T [49], Sequencer-S's performance is more sustainable. The relative accuracy is consistently better than ConvNeXt [49], which is influential in the lower-resolution band, and, at $448^2$ resolution, 0.6% higher than CycleMLP [7], which is influential in the double-resolution band. It is noteworthy that Sequencer continues to maintain high accuracy on double resolution.

The higher the input resolution, the higher memory-efficiency and throughput of Sequencers when compared to DeiT [72]. Figure 3 shows the efficiency of Sequencer2D-S when compared to DeiT-S and ConvNeXt-T [49]. Memory consumption increases rapidly in DeiT-S and ConvNeXt-T with increasing input resolution, but more gradual increase in Sequencer2D-S. The result strongly implies that it has more practical potential as the resolution increases than the ViTs. At a resolution of $224^2$, it is behind DeiT in throughput, but it stands ahead of DeiT when images with a resolution of $896^2$ are input.

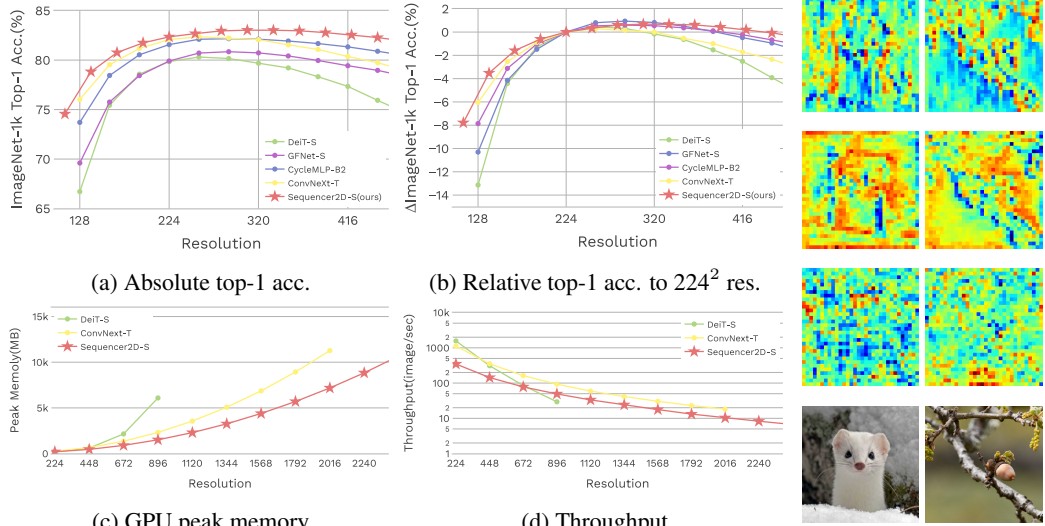

(a) Absolute top-1 acc.

(b) Relative top-1 acc. to $224^2$ res.

(c) GPU peak memory

(d) Throughput

Figure 3: **Top.** Resolution adaptability. Every model is trained at $224^2$ resolution and evaluated at various resolutions with no fine-tuning. **Bottom.** Comparisons among Sequencer2D-S, DeiT-S [72], and ConvNeXt-T [49] in (c) GPU peak memory and (d) throughput for different input image resolutions. Measured for each increment of $224^2$ resolution, points not plotted are when GPU memory is exhausted. The measurements are founded on a batch size of 16 and a single V100.

Figure 4: Part of states of the last BiLSTM2D layer in the Sequencer block of stage 1. From top to bottom: outputs of ver-LSTM, hor-LSTM, and ch-fusions and original images.

In general, CNNs have localized, layer-by-layer expanding receptive fields, and ViTs without shifted windows capture global dependencies, working the self-attention mechanism. In contrast, in the case of Sequencer, it is not clear how information is processed in Sequencer block. We calculated ERF [51] for ResNet-50 [22], DeiT-S [72], and Sequencer2D-S as shown in Figure 5. ERFs of Sequencer2D-S form a cruciform shape in all layers. The trend distinguishes it from well-known models such as DeiT-S and ResNet-50. More remarkably, in shallow layers, Sequencer2D-S has a wider ERF than ResNet-50, although not as wide as DeiT. This observation confirms that LSTMs in Sequencer can model long-term dependencies as expected and that Sequencer recognizes sufficiently long vertical or horizontal regions. Thus, it can be argued that Sequencer recognizes an image in a very different way than CNNs or ViTs. For more details on ERF and additional visualization, see Appendix D.

Moreover we also visualized a hidden state of vertical and horizontal BiLSTM, and a feature map after channel fusion, and the results are visualized in Figure 4. It demonstrates that our Sequencer has the hidden states interact with each other over the vertical and horizontal directions. The closer tokens are in position, the stronger their interaction tends to be; the farther tokens are in position, the less their interaction tends to be.

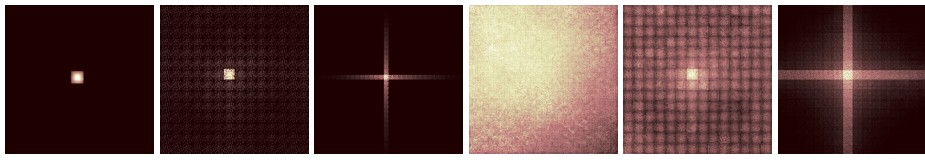

(a) RN50/first  (b) DeiTS/first  (c) SeqS/first  (d) RN50/last  (e) DeiTS/last  (f) SeqS/last

Figure 5: The visualizations are the ERFs of Sequencer2D-S and comparative models such as ResNet-50 and DeiT-S. The left of the slash denotes the model name, and the right of the slash denotes the location of the block of output used to generate the ERFs. The ERFs are rescaled from 0 to 1. The brighter and more influential the region is, the closer to 1, and the darker, the closer to 0.

## 5   Conclusions

We propose a novel and simple architecture that leverages LSTM for computer vision. It is demonstrated that new modeling with LSTM instead of the self-attention layer can achieve competitive performance with current state-of-the-art models. Our experiments show that Sequencer has a good memory-resource/accuracy and parameter/accuracy tradeoffs, comparable to the main existing methods. Despite the impact of recursion on throughput, we have demonstrated benefits over it. We believe that these results raise a number of interesting issues. Improving Sequencer's poor throughput is one example. Moreover, we expect that investigating the internal mechanisms of our model using methods other than ERF will further our understanding of how this architecture works. In addition, it would be important to analyze in more detail the features learned by Sequencer in comparison to other architectures. We hope this will lead to a better understanding of the role of various inductive biases in computer vision. Furthermore, we expect that our results trigger further study beyond the domain or research area. Especially, it would be a very interesting open question to see if such a design works with time-series data in vision such as video or in a multi-modal problem setting combined with another modality such as video with audio.

## Acknowledgments and Disclosure of Funding

Our colleagues at AnyTech Co., Ltd. provided valuable comments on the early versions and encouragement. We thank them for their cooperation. In particular, We thank Atsushi Fukuda for organizing discussion opportunities. We also thank people who support us, belonging to Graduate School of Artificial Intelligence and Science, Rikkyo University.

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
