# A Societal impact

The impact of this study on society has both positive and negative aspects. Here we discuss each.

On the positive side, our proposal would promote modeling methods using LSTMs in computer vision. This study takes image patches as tokens and models their relationships with LSTMs. Although LSTMs have been used in computer vision, designing image recognition with a module that includes LSTMs in the spatial direction as the main elements, as our study does, is new. It is exciting to see if this design benefits computer vision tasks other than image classification. Thus, our study would be an impetus for further research on its application to various computer vision tasks.

On the other side, our architecture may increase the carbon dioxide footprint: the study of new architectures for vision, such as Sequencer, requires iterative training of models for long periods to optimize the model's design. In particular, Sequencer is not a FLOPs-friendly design, and the amount of carbon dioxide emitted during training is likely to be high. Therefore, considering the environmental burden caused by the training of Sequencers, research to reduce the computational cost of Sequencers is also desired by society.

# B Implementation details

In this section, implementation details are supplemented. We describe the pseudocode of the BiLSTM2D layer, the architecture details, settings for training on IN-1K, and introduce settings for transfer learning.

## B.1 Pseudocode

---
**Algorithm 1** Pseudocode of BiLSTM2D layer.

---

```
# B: batch size H: height, W: width, C: channel, D: hidden dimension
# x: input tensor of shape (B, H, W, C)
### initialization ###
self.rnn_v = nn.LSTM(C, D, num_layers=1, batch_first=True, bias=True, bidirectional=True)
self.rnn_h = nn.LSTM(C, D, num_layers=1, batch_first=True, bias=True, bidirectional=True)
self.fc = nn.Linear(4 * D, C)
### forward ###
def forward(self, x):
    v, _ = self.rnn_v(x.permute(0, 2, 1, 3).reshape(-1, H, C))
    v = v.reshape(B, W, H, -1).permute(0, 2, 1, 3)
    h, _ = self.rnn_h(x.reshape(-1, W, C))
    h = h.reshape(B, H, W, -1)
    x = torch.cat([v, h], dim=-1)
    x = self.fc(x)
    return x
```

---

## B.2 Architecture details

This subsection describes Sequencer's architecture. The architectural details are shown in Table 4 and 5.

Sequencer2D-S is based on a ViP-S/7-like architecture. We intend to directly compare the BiLSTM2D layer in Sequencer2D, which has a similar structure, with the Permute-MLP layer in ViP-S/7. Table 4 is a summary of the architecture. In keeping with ViP, the first stage of Sequencers involves patch embedding with a 7x7 kernel. The second stage of Sequencers performs patch embedding with a 2x2 kernel, but the following two stages have no downsampling. The classifier of Sequencers then continues with layer normalization (LN) [1], followed by global average pooling and a linear layer. The number of blocks of Sequencer2D-S, Sequencer2D-M, and Sequencer2D-L correspond to ViP-S/7, ViP-M/7, and ViP-L/7, respectively. However, as described in Appendix C, we configure the dimension of the block to be different from ViP-M/7 and ViP-L/7 for Sequencer2D-M and Sequencer2D-L, respectively, because high dimension causes over-fitting.

VSequencer is a bit different from Sequencer2D in that it is non-hierarchical architecture. Table 5 define that no downsampling is performed in the second stage, instead of downsampling with 14x14 kernel for patch embedding in the first stage. In addition, we match the dimension of the blocks in the first stage to the dimension of the subsequent blocks.

Following the overall architecture, we describe the details of the modules not mentioned in the main text. Sequencer2D block and the Vanilla Sequencer block use LNs [1] for the normalization layers. We follow previous studies for the channel MLPs of these blocks and employ MLPs with Gaussian Error Linear Units (GELUs) [25] as the activation function; the ratio of increasing dimension in MLPs is uniformly 3x, as shown in Table 4 and 5.

### B.3 IN-1K settings

On IN-1K dataset [39], we utilize the hyper-parameters displayed in Table 6 to scratch train models in subsections 4.1 and 4.3. All Sequencer variants, including the models in the ablation study, follow almost the same settings for pre-training. However, the stochastic depth rate and batch size are adjusted depending on the model variant. The models in the ablation study are Sequencer2D-S based because of the following Sequencer2D-S settings.

The fine-tuning Sequencer2D-L↑ $392^2$ in subsection 4.2 has slightly different hyper-parameters than the pre-training models. There are changes in the settings for the number of epochs and learning rate because it uses trained weights, so there is no need to increase these hyper-parameters. In addition, we used crop ratio 0.875 during testing in the pre-training models instead of crop ratio 1.0 in the fine-tuning model.

### B.4 Transfer learning settings

Details of the datasets used for transfer learning in subsection 4.4 are shown in Table 7. This summary includes for each dataset CIFAR-10 [38], CIFAR-100 [38], Flowers-102 [55], and Stanford Cars [37], the number of training images, test images, and number of categories are listed.

Table 6 demonstrates the hyperparameters used in transfer learning with these datasets. The training epochs are especially adjusting to the datasets and changing them. The reason for this is attributable to the different sizes of the datasets.

## C  More results

This section discusses additional results that could not be addressed in the main text. The contents of the experiment consist of three parts: an evaluation of robustness in subsection C.1, an evaluation of generalization performance in subsection C.2, and a discussion of over-fitting in subsection C.3.

### C.1  Robustness

In this subsection, we evaluate the robustness of Sequencer. There are two main evaluation methods, benchmark datasets and adversarial attacks.

Evaluation with benchmark datasets reveals nice robustness of Sequencer. The evaluation results are summarized in Table 8. We test our models, trained on only IN-1K, on several datasets such as ImageNet-A/R/Sketch/C (IN-A/R/Sketch/C) [26, 23, 78, 24] to evaluate robustness. We evaluate our models on IN-C with mean corruption error (mCE), and on other datasets with top-1 accuracy. This result leads us to suggest that for models with a similar number of parameters, Sequencer is conquered by Swin and is robust enough to be competitive with ConvNeXt. Table 9 shows detail evaluation on IN-C. According to the results, it is understood that Sequencer is more immune to corruptions other than Noise than Swin and ConvNeXt, and, in particular, the model is less sensitive to weather conditions.

Sequencers are tolerant of principal adversarial attacks. We evaluate robustness using the single-step attack algorithm FGSM [18] and multi-step attack algorithm PGD [52]. Both algorithms give a perturbation of max magnitude 1. For PGD, we choose steps 5 and step size 0.5. This setup is based on RVT [53]. Table 9 indicates that Sequencer2D-L defeats in both FGSM and PGD compared to other models. Thus, Sequencer has an advantage over conventional models, such as RVT, which tout robustness on these adversarial attacks.

Table 4: **Variants of Sequencer2D and details**. "d" denotes the input/output dimension, and $D$ denotes the hidden dimension as above. "$\downarrow n$" (e.g., $\downarrow 2$) shows the stride of the downsampling is $n$

| | Sequencer2D-S | Sequencer2D-M | Sequencer2D-L |
|---|---|---|---|
| stage 1 | Patch Embedding$\downarrow 7$ 
 $\begin{bmatrix} \text{BiLSTM2D: 192d} \\ D = 48 \\ \text{MLP: 3 exp. ratio} \end{bmatrix} \times 4$ | Patch Embedding$\downarrow 7$ 
 $\begin{bmatrix} \text{BiLSTM2D: 192d} \\ D = 48 \\ \text{MLP: 3 exp. ratio} \end{bmatrix} \times 4$ | Patch Embedding$\downarrow 7$ 
 $\begin{bmatrix} \text{BiLSTM2D: 192d} \\ D = 48 \\ \text{MLP: 3 exp. ratio} \end{bmatrix} \times 8$ |
| stage 2 | Patch Embedding$\downarrow 2$ 
 $\begin{bmatrix} \text{BiLSTM2D: 384d} \\ D = 96 \\ \text{MLP: 3 exp. ratio} \end{bmatrix} \times 3$ | Patch Embedding$\downarrow 2$ 
 $\begin{bmatrix} \text{BiLSTM2D: 384d} \\ D = 96 \\ \text{MLP: 3 exp. ratio} \end{bmatrix} \times 3$ | Patch Embedding$\downarrow 2$ 
 $\begin{bmatrix} \text{BiLSTM2D: 384d} \\ D = 96 \\ \text{MLP: 3 exp. ratio} \end{bmatrix} \times 8$ |
| stage 3 | Point-wise Linear 
 $\begin{bmatrix} \text{BiLSTM2D: 384d} \\ D = 96 \\ \text{MLP: 3 exp. ratio} \end{bmatrix} \times 8$ | Point-wise Linear 
 $\begin{bmatrix} \text{BiLSTM2D: 384d} \\ D = 96 \\ \text{MLP: 3 exp. ratio} \end{bmatrix} \times 14$ | Point-wise Linear 
 $\begin{bmatrix} \text{BiLSTM2D: 384d} \\ D = 96 \\ \text{MLP: 3 exp. ratio} \end{bmatrix} \times 16$ |
| stage 4 | Point-wise Linear 
 $\begin{bmatrix} \text{BiLSTM2D: 384d} \\ D = 96 \\ \text{MLP: 3 exp. ratio} \end{bmatrix} \times 3$ | Point-wise Linear 
 $\begin{bmatrix} \text{BiLSTM2D: 384d} \\ D = 96 \\ \text{MLP: 3 exp. ratio} \end{bmatrix} \times 3$ | Point-wise Linear 
 $\begin{bmatrix} \text{BiLSTM2D: 384d} \\ D = 96 \\ \text{MLP: 3 exp. ratio} \end{bmatrix} \times 4$ |
| classifier | Layer Norm., Global Average Pooling, Linear | | |

## C.2   Generalization ability

The generalization ability of Sequencers is also impressive. We evaluate our models on ImageNet-Real/V2 (IN-Real/V2) [2, 62] to test their generalization performance: IN-Real is a re-labeled dataset of the IN-1K validation set, and IN-V2 is the dataset that re-collects the IN-1K validation set. Table 8 shows the results of evaluating the top-1 accuracy on both datasets. We reveal an understanding of the Sequencer's excellent generalization ability.

## C.3   Over-fitting

Wide Sequencers tend to be over-trained. We scratch-train Sequencer2D-Lx1.3, which has 4/3 times the dimension of each layer of Sequencer2D-L, on IN-1K. The training utilizes the same conditions as Sequencer2D-L. Consequently, as Table 10 shows, Sequencer2D-Lx1.3 has 0.8% less accuracy than Sequencer2D-L. Figure 6 illustrates the cross-entropy evolution and top-1 accuracy on IN-1K validation set for the two models. On the one hand, cross-entropy decreased on Sequencer2D-L in the last 100 epochs. On the other hand, Sequencer2D-Lx1.3 is increasing. Thus, widening Sequencer is counterproductive for training.

## C.4   Semantic segmentation

We evaluate models with Sequencer as the backbone for a semantic segmentation task. We trained and evaluated on ADE20K dataset [89], a well-known scene parsing benchmark. The dataset consists of the training set with about 20k images and the validation set with about 2k, covering 150 fine-grained semantic classes. We employed Sequencer as the backbone of SemanticFPN [36] to train and evaluate semantic segmentation. The training adopts a batch size of 32 and AdamW [50] with the initial learning rate of 2e-4, decay in the polynomial decay schedule with a power of 0.9, and 40k iterations

Table 5: **Variants of VSequencer and details**. "d" denotes the input/output dimension, and $D$ denotes the hidden dimension as above. "$\downarrow n$" (e.g., $\downarrow 2$) shows the stride of the downsampling is $n$

|  | VSequencer-S | | VSequencer-S(H) | | VSequencer-S(PE) | |
|---|---|---|---|---|---|---|
| stage 1 | Patch Embedding$\downarrow$14 | | Patch Embedding$\downarrow$7 | | Patch Embedding$\downarrow$14 | |
| | BiLSTM: 384d
$D = 192$
MLP: 3 exp. ratio | $\times 4$ | BiLSTM: 192d
$D = 96$
MLP: 3 exp. ratio | $\times 4$ | BiLSTM: 384d
$D = 192$
MLP: 3 exp. ratio | $\times 4$ |
| stage 2 | Point-wise Linear | | Patch Embedding$\downarrow$2 | | Point-wise Linear | |
| | BiLSTM: 384d
$D = 192$
MLP: 3 exp. ratio | $\times 3$ | BiLSTM: 384d
$D = 192$
MLP: 3 exp. ratio | $\times 3$ | BiLSTM: 384d
$D = 192$
MLP: 3 exp. ratio | $\times 3$ |
| stage 3 | Point-wise Linear | | Point-wise Linear | | Point-wise Linear | |
| | BiLSTM: 384d
$D = 192$
MLP: 3 exp. ratio | $\times 8$ | BiLSTM: 384d
$D = 192$
MLP: 3 exp. ratio | $\times 8$ | BiLSTM: 384d
$D = 192$
MLP: 3 exp. ratio | $\times 8$ |
| stage 4 | Point-wise Linear | | Point-wise Linear | | Point-wise Linear | |
| | BiLSTM: 384d
$D = 192$
MLP: 3 exp. ratio | $\times 3$ | BiLSTM: 384d
$D = 192$
MLP: 3 exp. ratio | $\times 3$ | BiLSTM: 384d
$D = 192$
MLP: 3 exp. ratio | $\times 3$ |
| classifier | Layer Norm., Global Average Pooling, Linear | | | | | |

(a) Cross entropy

(b) Top-1 accuracy

Figure 6: **Comparison of different model widths**. (a) is cross entropy, (b) is top-1 accuracy comparison, on IN-1K validation set. **The blue curve** represents the original Sequencer2D-L, which did not produce any problems and is learning all the way through. In contrast, **the green curve** represents the wider Sequencer2D-Lx1.3. This model stalls in the second half and is somewhat degenerate.

of training. These settings follow Metaformer [83]. Table 3 of the result indicates that Sequencer has the generalization for segmentation is comparable to other leading models.

## C.5 Object Detection

We evaluate Sequencer on COCO benchmark [45]. The dataset consists of 118k training images and 5k validation images. Sequencer with ImageNet pre-trained weights is employed as the backbone of RetinaNet [44]. Following [44], we employ AdamW, batch size of 16, and $1\times$ training schedule. Table 11 shows that Sequencer is not suited for existing standard object detection models such as

Table 6: **Hyper-parameters**. ↑ denotes fine-tuning pre-trained model on IN-1K. Multiple values are for each model, respectively.

| Training config. | Sequencer2D-S/M/L 224² | Sequencer2D-L↑ 392² | Sequencer2D-S↑/M↑/L↑ 224² |
|---|---|---|---|
| dataset | IN-1K [39] | IN-1K [39] | CIFAR$^{10, 100}$, Flowers, Cars |
| optimizer | AdamW [50] | AdamW [50] | AdamW [50] |
| base learning rate | 2e-3/1.5e-3/1e-3 | 5e-5 | 1e-4 |
| weight decay | 0.05 | 1e-8 | 1e-4 |
| optimizer $\epsilon$ | 1e-8 | 1e-8 | 1e-8 |
| optimizer momentum | $\beta_1, = 0.9, \beta_2$=0.999 | $\beta_1, = 0.9, \beta_2$=0.999 | $\beta_1, = 0.9, \beta_2$=0.999 |
| batch size | 2048/1536/1024 | 512 | 512 |
| training epochs | 300 | 30 | CIFAR: 200, Others: 1000 |
| learning rate schedule | cosine decay | cosine decay | cosine decay |
| lower learning rate bound | 1e-6 | 1e-6 | 1e-6 |
| warmup epochs | 20 | None | 5 |
| warmup schedule | linear | None | linear |
| warmup learning rate | 1e-6 | None | 1e-6 |
| cooldown epochs | 10 | None | 10 |
| crop ratio | 0.875 | 1.0 | 0.875 |
| randaugment [11] | (9, 0.5) | (9, 0.5) | (9, 0.5) |
| mixup $\alpha$ [87] | 0.8 | 0.8 | 0.8 |
| cutmix $\alpha$ [85] | 1.0 | 1.0 | 1.0 |
| random erasing [88] | 0.25 | 0.25 | None |
| label smoothing [66] | 0.1 | 0.1 | 0.1 |
| stochastic depth [30] | 0.1/0.2/0.4 | 0.4 | 0.1/0.2/0.4 |
| gradient clip | None | None | 1 |

Table 7: **Transfer learning datasets**.

| Dataset | Train Size | Test size | #Classes |
|---|---|---|---|
| CIFAR-10 [38] | 50,000 | 10,000 | 10 |
| CIFAR-100 [38] | 50,000 | 10,000 | 100 |
| Flowers-102 [55] | 2,040 | 6,149 | 102 |
| Stanford Cars [37] | 8,144 | 8,041 | 196 |

RetinaNet. It shows no improvement trend for model scaling. It also struggles to detect small objects, making RNN-based object detection models an issue to consider in the future.

### C.6 More studies

**Method of merge** As shown in Figure 2, "`concatenate`" is used to merge the vertical BiLSTM and horizontal BiLSTM outputs but "`add`" can also be used. See Table 12a for the result of the experiment.

## D Effective receptive field

This section covers in detail the effective receptive fields (ERFs) [51] used in the visualization in subsection 4.5. First, we explain how the visualized effective receptive fields are obtained. Second, we present other visualization results not addressed in the main text. The ERF's calculations in this paper are based on [14].

### D.1 Calculation of visualized ERFs

The ERF [51] is a technique for calculating the pixels that contribute to the center of a output feature maps of a neural network. Let $\mathbf{I} \in \mathbb{R}^{n \times h \times w \times c}$ be a input image collection and $\mathbf{O} \in \mathbb{R}^{n \times h' \times w' \times c'}$ be the output feature map collection. The center of the output feature map can be expressed as $\mathbf{O}_{:, \lfloor h'/2 \rfloor, \lfloor w'/2 \rfloor, :}$, where $\lfloor \cdot \rfloor$ is the floor function. Each element of the derivative of $\mathbf{O}_{i, \lfloor h'/2 \rfloor, \lfloor w'/2 \rfloor, j}$ to $\mathbf{I}$, i.e., $\frac{\partial \left( \sum_{i,j} \mathbf{O}_{i, \lfloor h'/2 \rfloor, \lfloor w'/2 \rfloor, j} \right)}{\partial \mathbf{I}}$, represents to what extent the center of the output feature map

Table 8: **The robustness** is evaluated on IN-A [26] (top-1 accuracy), IN-R [23] (top-1 accuracy), IN-Sketch [78] (top-1 accuracy), IN-C [24] (mCE), FGSM [18] (top-1 accuracy), and PGD [52] (top-1 accuracy). **The generalization ability** is evaluated on IN-Real [2] and IN-V2 [62]. We denote the higher as better value as ↑ and the lower as better value as ↓. Rather than those reported in the original paper, the values we observed are marked with †. If the model name has †, it means that we observed all the metrics of the model.

| Model | #Param. | FLOPs | Clean(↑) | A(↑) | R(↑) | Sk.(↑) | C(↓) | FGSM(↑) | PGD(↑) | Real(↑) | V2(↑) |
|---|---|---|---|---|---|---|---|---|---|---|---|
| Swin-T [48] | 28M | 4.5G | 81.2 | 21.6 | 41.3 | 29.1 | 62.0 | 33.7 | 7.3 | 86.7† | 69.6† |
| ConvNeXt-T [49] | 29M | 4.5G | 82.1 | 24.2 | 47.2 | 33.8 | 53.2 | 37.8† | 10.5† | 87.3† | 71.0† |
| RVT-S* [53] | 23M | 4.7G | 81.9 | 25.7 | 47.7 | 34.7 | 49.4 | 51.8 | 28.2 | - | - |
| Sequencer2D-S | 28M | 8.4G | 82.3 | 26.7 | 45.1 | 33.4 | 53.0 | 49.2 | 25.0 | 87.4 | 71.8 |
| Sequencer2D-M | 38M | 11.1G | 82.8 | 30.5 | 46.3 | 34.7 | 51.8 | 50.8 | 26.3 | 87.6 | 72.5 |
| Swin-S [48]† | 50M | 8.7G | 83.2 | 32.5 | 45.2 | 32.3 | 54.9 | 45.9 | 18.1 | 87.7 | 72.1 |
| ConvNeXt-S† [49] | 50M | 8.7G | 83.1 | 31.3 | **49.6** | **37.1** | 49.5 | 46.1 | 17.7 | **88.1** | 72.5 |
| Sequencer2D-L | 54M | 16.6G | **83.4** | **35.5** | 48.1 | 35.8 | **48.9** | **53.1** | **30.9** | 87.9 | **73.4** |
| Swin-B [48] | 88M | 15.4G | 83.4 | 35.8 | 46.6 | 32.4 | 54.4 | 49.2 | 21.3 | 89.2† | 75.6† |
| ConvNeXt-B [49] | 89M | 15.4G | 83.8 | 36.7 | 51.3 | 38.2 | 46.8 | 47.5† | 18.3† | 88.4† | 73.7† |
| RVT-B* [53] | 92M | 17.7G | 82.6 | 28.5 | 48.7 | 36.0 | 46.8 | 53.0 | 29.9 | - | - |

Table 9: Details of robustness evaluation with IN-C.

| Model | mCE | Noise | | | Blur | | | | Weather | | | | Digital | | | |
|---|---|---|---|---|---|---|---|---|---|---|---|---|---|---|---|---|
| | | Gauss. | Shot | Impulse | Defocus | Glass | Motion | Zoom | Snow | Frost | Fog | Bright | Contrast | Elastic | Pixel | JPEG |
| Swin-S [48] | 54.9 | 42.9 | 44.9 | 43.2 | 61.3 | 74.1 | 56.6 | 67.5 | 50.8 | 48.5 | 46.0 | 44.1 | 42.1 | 68.9 | 62.1 | 70.7 |
| ConvNeXt-S [49] | 49.5 | **38.1** | **39.1** | **37.9** | 57.8 | 72.5 | **51.8** | **61.9** | 46.1 | 43.8 | 44.6 | 39.6 | 37.6 | 66.7 | 55.1 | **50.1** |
| Sequencer2D-L | **48.9** | 43.3 | 42.0 | 41.4 | **55.2** | **71.0** | **51.8** | 63.3 | **44.2** | **41.0** | **41.9** | **37.1** | **33.8** | **66.6** | **50.4** | 51.1 |

changes for each perturbation of each pixel in each input image. Adding these together for all images and channels, we can calculate the average pixel contribution for all input images, which can be activated with a Rectified Linear Unit (ReLU) to get the positively contributing pixel values $\mathbf{P} \in \mathbb{R}^{n \times h \times w \times c}$, defined by

$$\mathbf{P} = \text{ReLU}\left( \frac{\partial \left( \sum_{i,j} \mathbf{O}_{i,\lfloor h'/2,\rfloor \lfloor w'/2 \rfloor,j} \right)}{\partial \mathbf{I}} \right) . \tag{8}$$

Furthermore, the score $\mathrm{S} \in \mathbb{R}^{h \times w}$ is calculated by

$$\mathrm{S} = \log_{10}\left( \sum_{i,j} \mathbf{P}_{i,:,:,j} + 1 \right) , \tag{9}$$

and S is called the effective receptive field.

Next, define a visualized effective receptive field based on the effective receptive field. We want to compare the effective receptive fields across models. We, therefore, calculate the score $\mathrm{S}_{\text{model}}$ for each model and rescale $\mathrm{S}_{\text{model}}$ from 0 to 1 across the models. The tensor calculated in this way is called the visualized effective receptive field.

The derivatives used in these definitions are efficient if they take advantage of the auto-grad mechanism. Indeed, we also relied on the automatic auto-grad function on `PyTorch` [56] to calculate the effective receptive fields.

Table 10: **Comparison of accuracy for different model widths**.

| Model | #Params. | FLOPs | Acc. |
|---|---|---|---|
| Sequencer2D-L | 54M | 16.6G | 83.4 |
| Sequencer2D-Lx1.3 | 96M | 29.4G | 83.0 |

Table 11: Object detection results on COCO dataset [45]

| Backbone | Params (M) | AP | $AP_{50}$ | $AP_{75}$ | $AP_S$ | $AP_M$ | $AP_L$ |
|---|---|---|---|---|---|---|---|
| ResNet-18 [22] | 21.3 | 31.8 | 49.6 | 33.6 | 16.3 | 34.3 | 43.2 |
| PoolFormer-S12 [83] | 21.7 | 36.2 | 56.2 | 38.2 | 20.8 | 39.1 | 48.0 |
| Sequencer2D-S | 37.3 | 33.6 | 54.8 | 34.8 | 15.3 | 37.5 | 50.2 |
| ResNet-50 [22] | 37.7 | 36.3 | 55.3 | 38.6 | 19.3 | 40.0 | 48.8 |
| PoolFormer-S24 [83] | 31.1 | 38.9 | 59.7 | 41.3 | 23.3 | 42.1 | 51.8 |
| Sequencer2D-M | 47.9 | 34.5 | 55.5 | 35.9 | 15.0 | 39.0 | 51.6 |
| ResNet-101 [22] | 56.7 | 38.5 | 57.8 | 41.2 | 21.4 | 42.6 | 51.1 |
| PoolFormer-S36 [83] | 40.6 | 39.5 | 60.5 | 41.8 | 22.5 | 42.9 | 52.4 |
| Sequencer2D-L | 63.9 | 35.0 | 56.4 | 36.5 | 16.5 | 39.6 | 51.6 |

Table 12: **More Sequencer ablation experiments**.

(a) Method of merge

| Union | #Params. | FLOPs | Acc. |
|---|---|---|---|
| add | 27M | 8.0G | 82.2 |
| concatnate | 28M | 8.4G | **82.3** |

## D.2 More visualization of ERFs

We introduce additional visualization and concrete visualization method. We experiment with visualization using input images of two different resolutions.

We visualize the effective receptive fields of Sequencer2D-S and comparative models by using $224^2$ resolution images. The method is applied to the following models for comparing: ResNet-50 [22], ConvNeXt-T [49], CycleMLP-B2 [7], DeiT-S [72], Swin-T [48], GFNet-S [61], and ViP-S/7 [28]. The object to be visualized is the output for each block, and the effective receptive fields are calculated. For example, in the case of Sequencer2D-S, the effective receptive fields are calculated for the output of each Sequencer block. We are rescaling within a value between 0 and 1 for the whole to effective receptive fields for each model block.

The effective receptive fields of Sequencer2D-S and comparative models are then visualized using input images with a resolution of $448^2$. The reason for running experiments is to verify how the receptive field is affected when the input resolution is increased compared to the $224^2$ resolution input image. Sequencer2D-S compare with ResNet-50 [22], ConvNeXt-T [49], CycleMLP-B2 [7], DeiT-S [72], and GFNet-S [28]. The method of visualization of the effective receptive field follows the case of input images with a resolution of $224^2$.

Sequencer has very distinctive cruciform ERFs in all layers. Table 7, 8, 9, 10, 11, 12, 13, and 14 illustrates this fact for $224^2$ resolution input images. Furthermore, as shown in Table 15, 16, 17, 18 and 19, we observe the same trend when the double resolution. The ERFs are structurally quite different from the ERFs other than ViP, which have a similar structure. ViP's ERFs have, on average, some also coverage except for the cruciforms. In contrast, Sequencer's ERFs are limited to the cruciform and its neighborhood.

It is interesting to note that Sequencer, with its characteristic ERFs, achieves high accuracy. It will be helpful for future architecture development because of the possibility of creating Sequencer-like ERFs outside of LSTM.

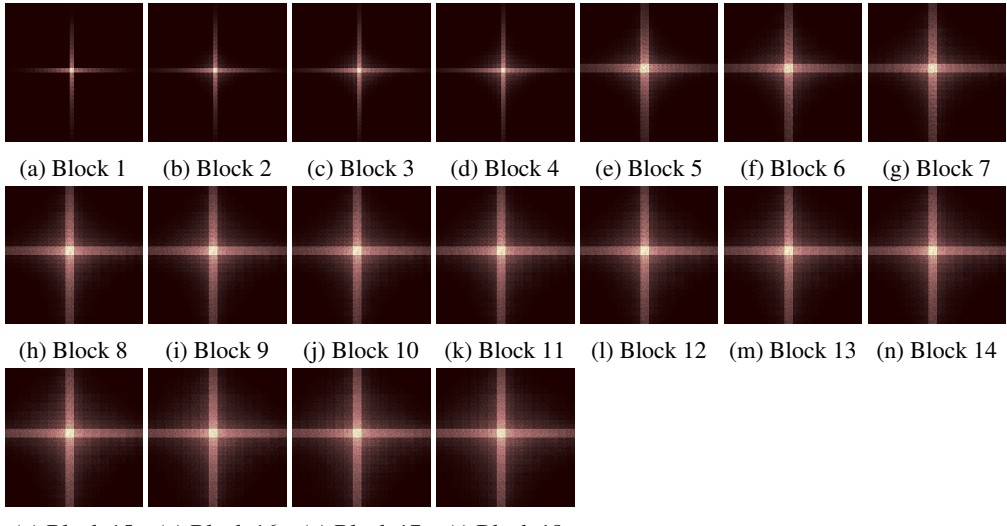

(a) Block 1 (b) Block 2 (c) Block 3 (d) Block 4 (e) Block 5 (f) Block 6 (g) Block 7

(h) Block 8 (i) Block 9 (j) Block 10 (k) Block 11 (l) Block 12 (m) Block 13 (n) Block 14

(o) Block 15 (p) Block 16 (q) Block 17 (r) Block 18

Figure 7: ERFs in Sequencer2D-S on images with resolution $224^2$.

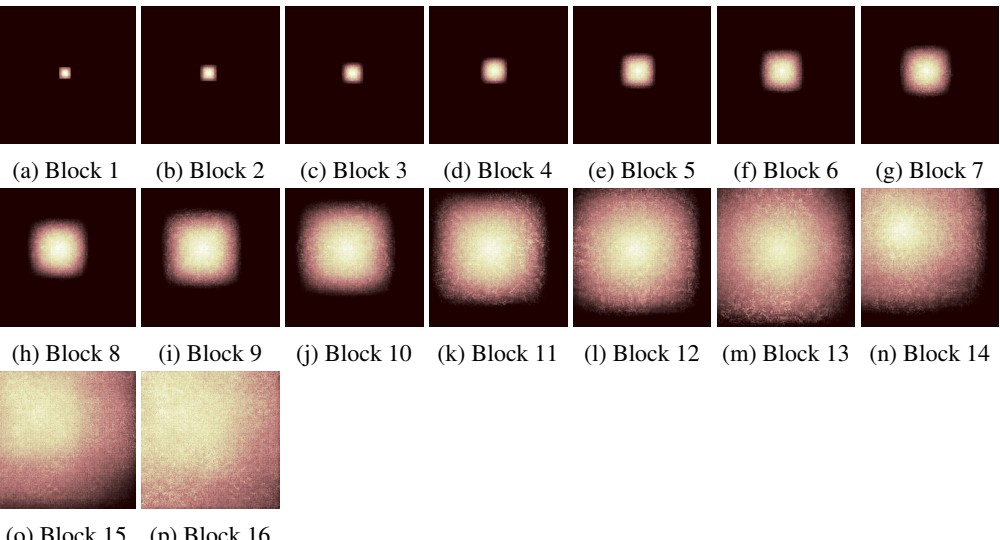

(a) Block 1 (b) Block 2 (c) Block 3 (d) Block 4 (e) Block 5 (f) Block 6 (g) Block 7

(h) Block 8 (i) Block 9 (j) Block 10 (k) Block 11 (l) Block 12 (m) Block 13 (n) Block 14

(o) Block 15 (p) Block 16

Figure 8: ERFs in ResNet-50 [22] on images with resolution $224^2$.

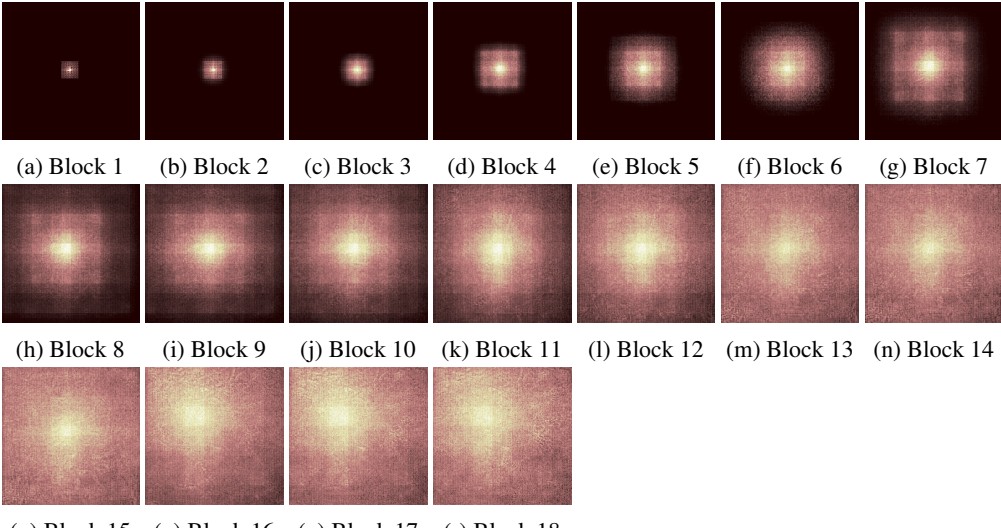

(a) Block 1    (b) Block 2    (c) Block 3    (d) Block 4    (e) Block 5    (f) Block 6    (g) Block 7

(h) Block 8    (i) Block 9    (j) Block 10    (k) Block 11    (l) Block 12    (m) Block 13    (n) Block 14

(o) Block 15    (p) Block 16    (q) Block 17    (r) Block 18

Figure 9: ERFs in ConvNeXt-T [49] on images with resolution $224^2$.

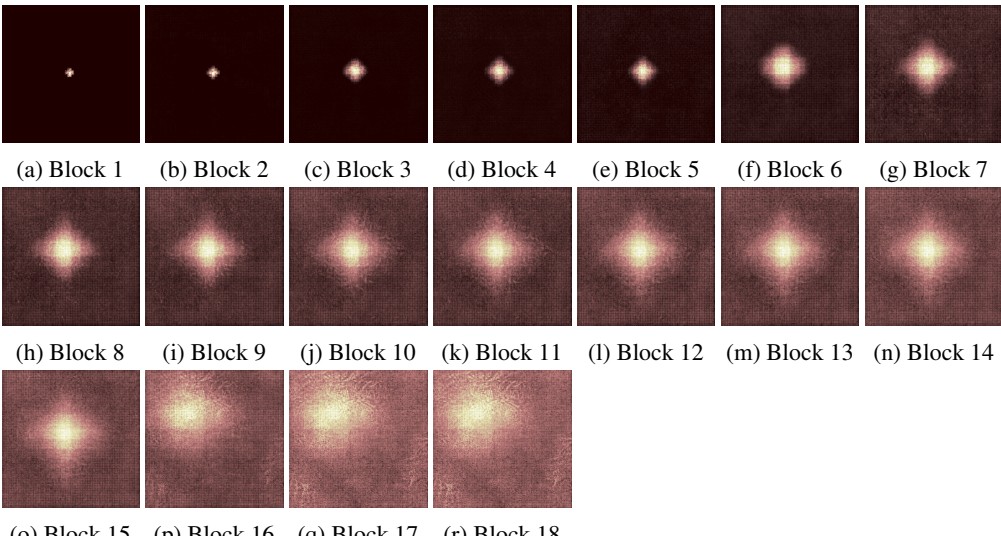

(a) Block 1    (b) Block 2    (c) Block 3    (d) Block 4    (e) Block 5    (f) Block 6    (g) Block 7

(h) Block 8    (i) Block 9    (j) Block 10    (k) Block 11    (l) Block 12    (m) Block 13    (n) Block 14

(o) Block 15    (p) Block 16    (q) Block 17    (r) Block 18

Figure 10: ERFs in CycleMLP-B2 [7] on images with resolution $224^2$.

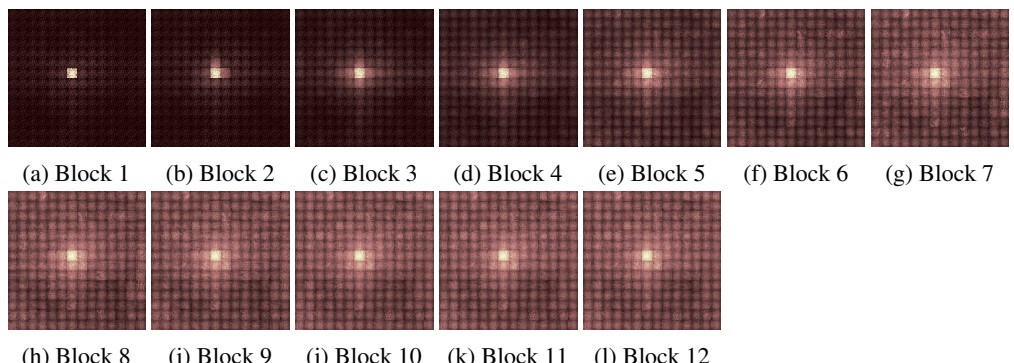

(a) Block 1    (b) Block 2    (c) Block 3    (d) Block 4    (e) Block 5    (f) Block 6    (g) Block 7

(h) Block 8    (i) Block 9    (j) Block 10    (k) Block 11    (l) Block 12

Figure 11: ERFs in DeiT-S [72] on images with resolution $224^2$.

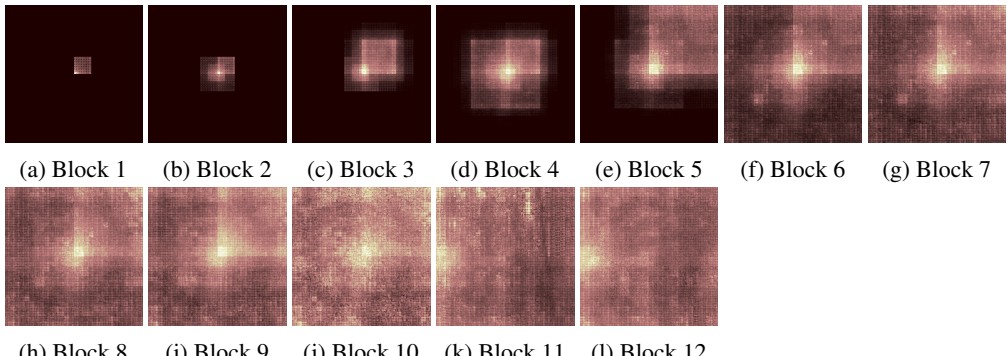

(a) Block 1    (b) Block 2    (c) Block 3    (d) Block 4    (e) Block 5    (f) Block 6    (g) Block 7

(h) Block 8    (i) Block 9    (j) Block 10    (k) Block 11    (l) Block 12

Figure 12: ERFs in Swin-T [48] on images with resolution $224^2$.

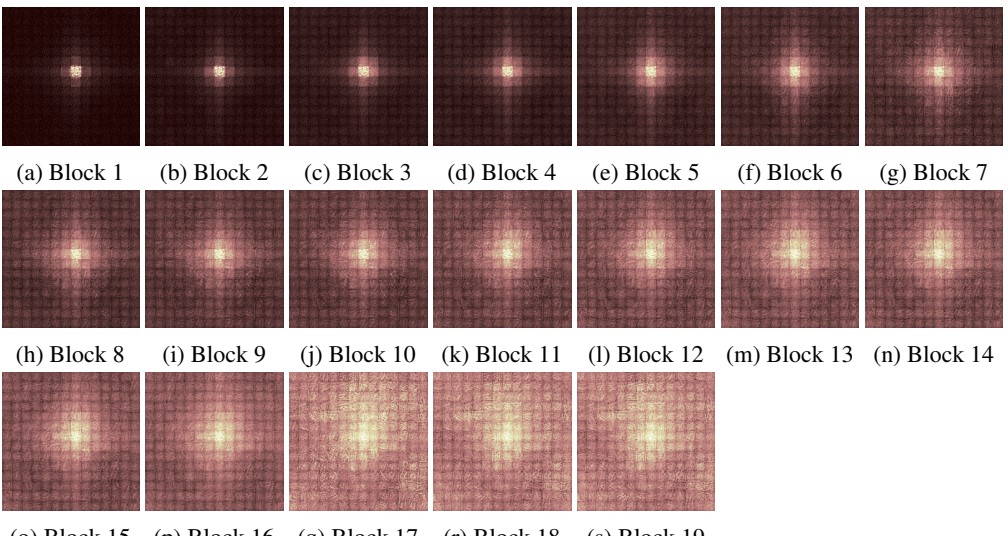

(a) Block 1    (b) Block 2    (c) Block 3    (d) Block 4    (e) Block 5    (f) Block 6    (g) Block 7

(h) Block 8    (i) Block 9    (j) Block 10    (k) Block 11    (l) Block 12    (m) Block 13    (n) Block 14

(o) Block 15    (p) Block 16    (q) Block 17    (r) Block 18    (s) Block 19

Figure 13: ERFs in GFNet-S [61] on images with resolution $224^2$.

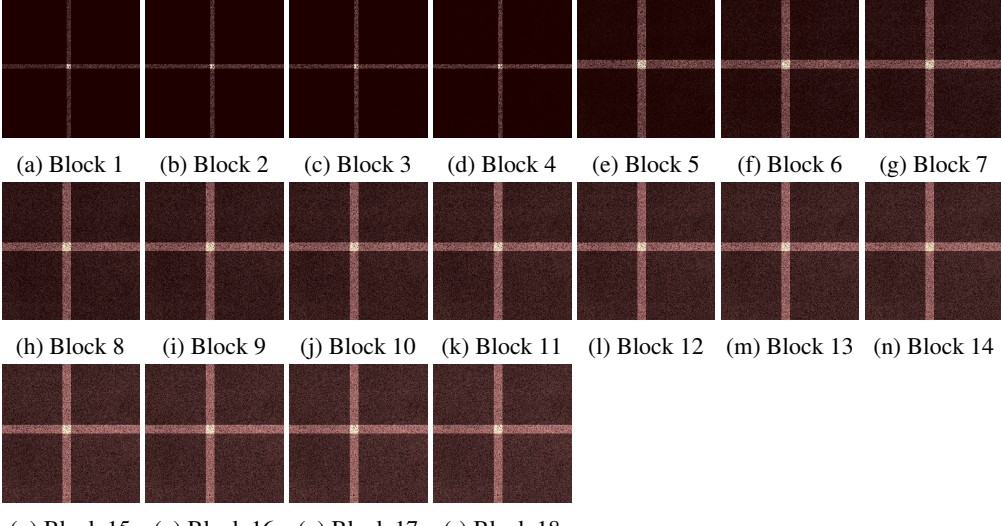

(a) Block 1    (b) Block 2    (c) Block 3    (d) Block 4    (e) Block 5    (f) Block 6    (g) Block 7

(h) Block 8    (i) Block 9    (j) Block 10    (k) Block 11    (l) Block 12    (m) Block 13    (n) Block 14

(o) Block 15    (p) Block 16    (q) Block 17    (r) Block 18

Figure 14: ERFs in ViP-S/7 [28] on images with resolution $224^2$.

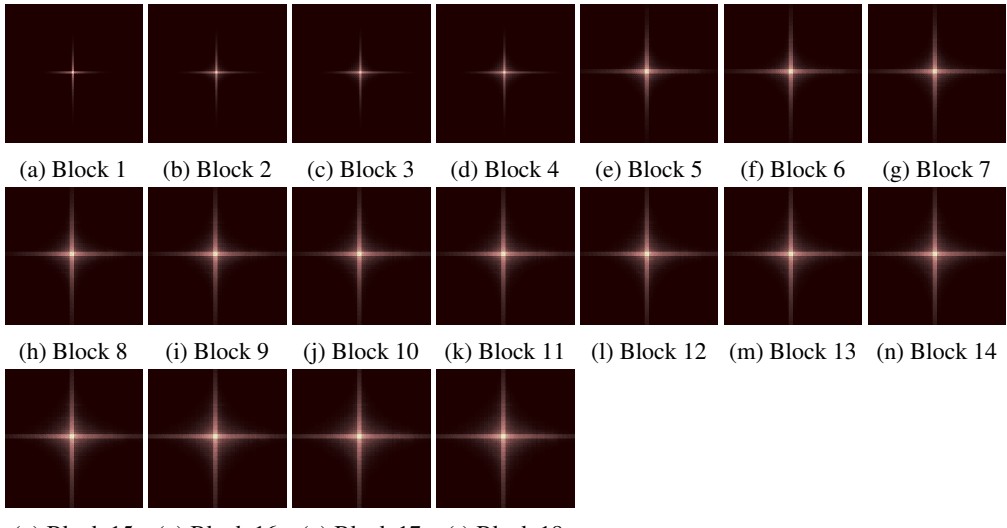

| (a) Block 1 | (b) Block 2 | (c) Block 3 | (d) Block 4 | (e) Block 5 | (f) Block 6 | (g) Block 7 |

| (h) Block 8 | (i) Block 9 | (j) Block 10 | (k) Block 11 | (l) Block 12 | (m) Block 13 | (n) Block 14 |

| (o) Block 15 | (p) Block 16 | (q) Block 17 | (r) Block 18 |

Figure 15: ERFs in Sequencer2D-S on images with resolution $448^2$.

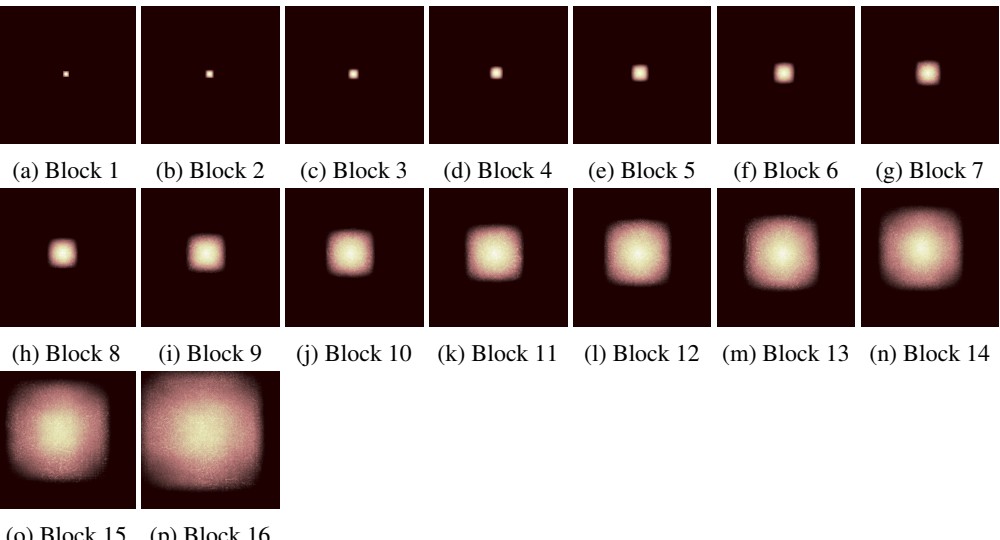

| (a) Block 1 | (b) Block 2 | (c) Block 3 | (d) Block 4 | (e) Block 5 | (f) Block 6 | (g) Block 7 |

| (h) Block 8 | (i) Block 9 | (j) Block 10 | (k) Block 11 | (l) Block 12 | (m) Block 13 | (n) Block 14 |

| (o) Block 15 | (p) Block 16 |

Figure 16: ERFs in ResNet-50 [22] on images with resolution $448^2$.

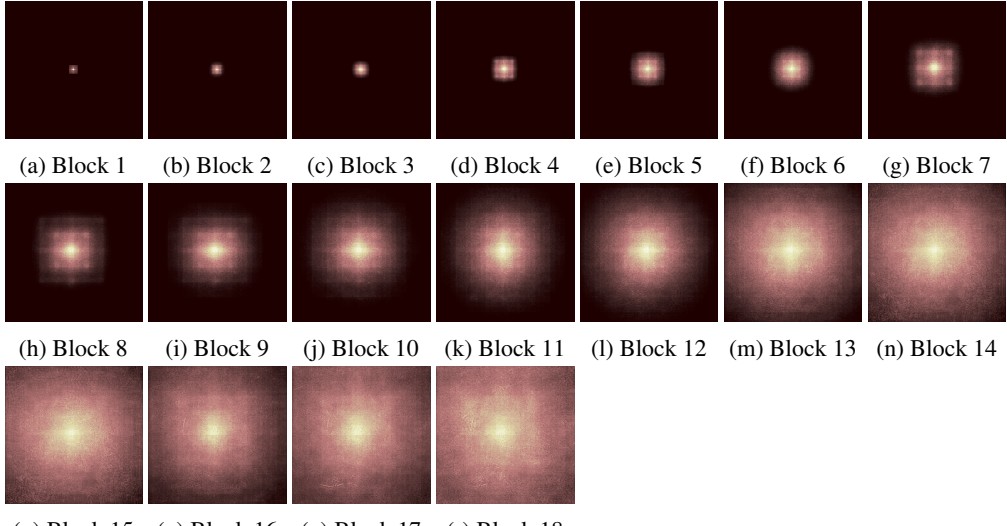

(a) Block 1    (b) Block 2    (c) Block 3    (d) Block 4    (e) Block 5    (f) Block 6    (g) Block 7

(h) Block 8    (i) Block 9    (j) Block 10    (k) Block 11    (l) Block 12    (m) Block 13    (n) Block 14

(o) Block 15    (p) Block 16    (q) Block 17    (r) Block 18

Figure 17: ERFs in ConvNeXt-T [49] on images with resolution $448^2$.

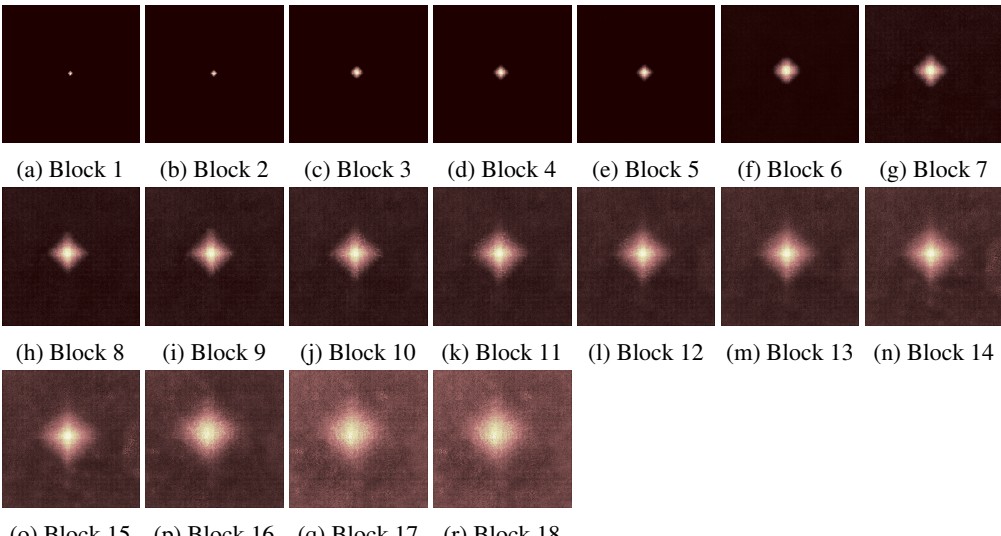

(a) Block 1    (b) Block 2    (c) Block 3    (d) Block 4    (e) Block 5    (f) Block 6    (g) Block 7

(h) Block 8    (i) Block 9    (j) Block 10    (k) Block 11    (l) Block 12    (m) Block 13    (n) Block 14

(o) Block 15    (p) Block 16    (q) Block 17    (r) Block 18

Figure 18: ERFs in CycleMLP-B2 [7] on images with resolution $448^2$.

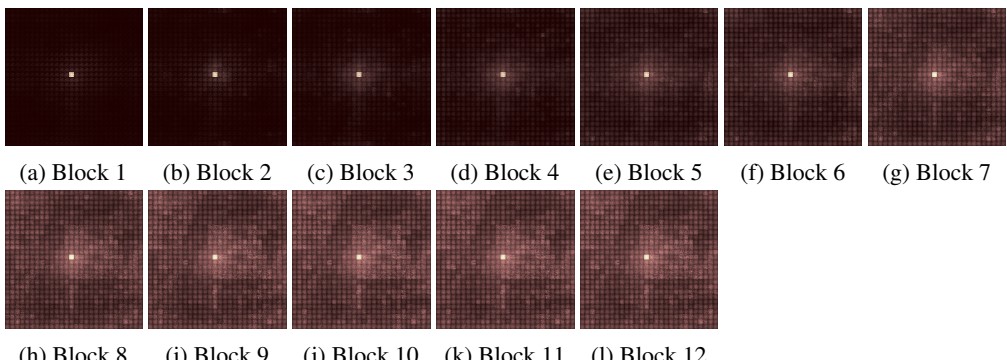

(a) Block 1    (b) Block 2    (c) Block 3    (d) Block 4    (e) Block 5    (f) Block 6    (g) Block 7

(h) Block 8    (i) Block 9    (j) Block 10    (k) Block 11    (l) Block 12

Figure 19: ERFs in DeiT-S [72] on images with resolution $448^2$.