# OpenReview forum: "Sequencer: Deep LSTM for Image Classification"
_NeurIPS.cc/2022/Conference — NeurIPS 2022 Accept_

### Official Review · Reviewer_ivfp · 2022-07-05

**Rating:** 6
**Confidence:** 4
**Soundness:** 3 good
**Presentation:** 4 excellent
**Contribution:** 3 good

**Summary:**

This paper proposed Sequencer by using deep LSTMs instead of self-attention for image classification. And many related works were compared in experiments to validate the performance of sequencer.

**Questions:**

1.	From Table 1, compared with ConvNetXt, Sequencer achieved worse performance at FLOPS and throughput with 0.2~0.3 top-1 accuracy increase. Meanwhile, the fine-tuned sequencer is worse than the fine-tuned ConvNetXt.
2.	In Figure 3, the performance between Sequencer and ConvNetXt seems to conflict with Table 1？

**Limitations:**

Some experimental results are not clearly explained.

**Strengths And Weaknesses:**

Strength: This paper proposed Sequencer, which uses LSTM instead of the self-attention for sequence modeling. This paper also proposed a two-dimensional version of Sequencer module, where an LSTM is decomposed into vertical and horizontal LSTMs to enhance performance. Experiments showed the advantages of Sequencers compared to the self-attention mechanism in transferability and resolution adaptability. The work is clearly stated, and the manuscript is well written.
Weakness: Some experimental results are not clearly explained.

---

> ### Author Response · Authors · 2022-08-02
> **Response to Reviewer ivfp**
>
> Thanks for your positive comments and questions on our paper. We would be happy to resolve your questions.
>
> ### Reply to Question 1
>
> We think our lack of representation has caused you to misunderstand our claim. The original Table 1 was misleading, thus we have annexed the accuracy before fine-tuning in ther revised version of Table 1. We have also added a supplement to its caption. Sequencer-L trained from scratch, is compared with ConvNeXt-"S" with comparable parameters. In contrast, fine-tuned Seuqnecer-L is compared with ConvNeXt-"B" which has more parameters than Seuqnecer-L. We reluctantly have postd ConvNeXt-"B" since [1] do not fine-tune ConvNeXt-"S" but ConvNeXt-"B". Non-finetuned ConvNeXt-"B" achieves 83.8% accuracy as reported in [1], 0.4% more than Sequencer-L. This suggests that the report, finetuned ConvNeXt-"B" is 0.5% more accurate than finetuned Seuqnecer-L, is not unnatural.
>
> ### Reply to Question 2
>
> We assume that you have misunderstood the model to which this experiment is to be compared due to the imperfections in our explanation. The only mention of which type of model is being compared  has been in Figure 3a and 3b, so we have added it to the text to reduce misunderstandings in the revised version. Since We could not figure out which part is inconsistent, we will explain Figure 3a and 3b for the moment. If you could add any additional information on the content that you feel is inconsistent, we would be glad to correct that part of our paper. Figure 3a and 3b compare the top-1 accuracy of DeiT-S, GFNet-S, CycleMLP-B2, Seuqnecer-S, and ConvNeXt-T models trained at a resolution of $224^2$, with different input image resolutions of ImageNet validation set during inference, with no fine-tuning. Figure 3a concisely plots the accuracy. For example, the accuracy of Seuqnecer-S and ConvNeXt-T are achieved as 82.3 and 82.1 in Table 1, respectively, so they are plotted at 82.3 and 82.1 on the line of resolution $224^2$. Figure 3b is relative to the accuracy at resolution $224^2$. Thus, any models are plotted $0$ on the $224^2$ resolution line. The table compares how much the inference accuracy drops as the resolution changes without being distracted by the difference in accuracy at resolution $224^2$.
>
> #### Reference
>
> [1] "A convnet for the 2020s." CVPR 2022

---

> > ### Comment · Reviewer_ivfp · 2022-08-09
> > **Thanks for the authors’ reply.**
> >
> > The authors have addressed my questions. I have no further question.

---

### Official Review · Reviewer_GTX2 · 2022-07-10

**Rating:** 6
**Confidence:** 4
**Soundness:** 3 good
**Presentation:** 3 good
**Contribution:** 3 good

**Summary:**

This paper proposes a new Sequencer architecture that replaces self-attention in ViT with BiLSTM(2D) for the image classification task. On ImageNet-1K dataset, Sequencer achieves better performance than current other similar scale models. The authors also show Sequencer is more robust to resolution variation and suffers from less severe accuracy degradation when the input resolution is increased.

**Questions:**

1. What is the running time of Sequencer and other baselines on ImageNet-1K? Are there any potential ways to reduce the computational cost, such as reducing the length of the sequence by downsampling? If so, can the authors report the model performance compared with the baselines under a similar scale of FLOPs and throughput?

2. See (2) of Cons. Can the authors provide more discussions and potential visualization on why Sequencer block is effective?

3. What if replace LSTM with other RNN cells, such as GRU? Will the model still work well? If not, which part is essential to the model performance?

**Limitations:**

The authors have discussed the limitations in the conclusion. Actually, it would be better if the authors can test the model's effectiveness on tasks that require sequence modeling such as video action recognition in the main paper.

**Strengths And Weaknesses:**

Pros:
1. This paper makes an attempt to use LSTM, an unexplored inductive bias, to replace self-attention in ViT for image classification and shows its effectiveness. This line of research helps the community understand what is indeed essential for vision tasks.

2. Strong results and extensive experiments. It compares with a series of related works based on various inductive biases and shows that it has superior performance and transferability under a similar scale of parameters. Besides, ablation studies are conducted.

Cons:
1. The computational cost is too high. As shown in Table 1, under a similar scale of model parameters, Sequencer usually needs 2x FLOPs and is 2x~10x lower throughput compared with other methods. Although this is not surprising due to the recursion in LSTM, I am still concerned about the practicality of this model with such a high computational cost.

2. Lack of reasoning on how using LSTM captures the spatial information and why it is so effective. In BiLSTM2D, it uses LSTM to capture dependencies from horizontal and vertical patches respectively. From my point of view, this design should not be as effective as global dependencies in self-attention since you may need to involve patches that are not necessarily in the same horizontal and vertical line to understand the objects in the images. Besides, I am also curious about what role the memory in LSTM plays in processing spatial information. The above analysis is critical for readers to understand the model but is missing in the paper.

---

> ### Author Response · Authors · 2022-08-02
> **Response to Reviewer GTX2 (1/2)**
>
>
> We appreciate your thought-provoking feedback and positive assessment. We have reflected them in the revised paper. The revised paper has been uploaded to OpenReview.
>
> ### Cons. 1
>
> > Sequencer usually needs 2x FLOPs and is 2x~10x lower throughput compared with other methods.
>
> We listed the throughput value for Sequencer2D-L incorrectly. The revised version corrects that. In the case of this correct result, Sequencer is 2x~7x lower throughput.
>
> > Although this is not surprising due to the recursion in LSTM, I am still concerned about the practicality of this model with such a high computational cost.
>
> The poor throughput is less notable at resolutions higher than 224x224 compared to other methods (Figure 3d). For example semantic segmentation often uses images with resolutions higher than 224x224, such as 512x512. In revised version, we have added the result showing that Sequencer is competitive with Poolformer in semantic segmentation task (See Section 4.4 and Appendix C.4).
>
> ### Reply to Question 1
>
> > What is the running time of Sequencer and other baselines on ImageNet-1K?
>
> The other baselines values are based on the cited papers, so we do not have the results at hand comparing ImageNet-1K training times. We do, however, measure training throughput for each baseline model. We provide the results:
>
> | Model | Infer Throughput(image/s) | Infer Peak Mem. | Train Throughput(image/s) | Train Peak Mem. |
> | --- | --- | --- | --- | --- |
> | RegNetY-4GF | 823 | 225 | 228 | 1136 |
> | ConvNeXt-T | 1124 | 248 | 337 | 1418 |
> | DeiT-S | 1569 | 180 | 480 | 1195 |
> | Swin-T | 894 | 308 | 268 | 1613 |
> | ViP-S/7 | 702 | 195 | 214 | 1587 |
> | CycleMLP-B2 | 586 | 234 | 158 | 1357 |
> | PoolFormer-S24 | 988 | 183 | 313 | 1461 |
> | Sequencer2D-S (Ours) | 347 | 196 | 110 | 1799 |
> | RegNetY-8GF | 751 | 333 | 211 | 1776 |
> | T2T-ViT$_{t}$-19 | 654 | 1140 | 197 | 3520 |
> | CycleMLP-B3 | 367 | 287 | 100 | 2326 |
> | PoolFormer-S36 | 673 | 220 | 213 | 2187 |
> | GFNet-H-S | 755 | 282 | 227 | 1740 |
> | Sequencer2D-M (Ours) | 270 | 244 | 83 | 2311 |
> | RegNetY-12GF | 695 | 440 | 199 | 2181 |
> | ConvNeXt-S | 717 | 341 | 212 | 2265 |
> | Swin-S | 566 | 390 | 165 | 2635 |
> | Mixer-B/16 | 1011 | 407 | 338 | 1864 |
> | ViP-M/7 | 395 | 396 | 130 | 3095 |
> | CycleMLP-B4 | 259 | 338 | 70 | 3272 |
> | PoolFormer-M36 | 496 | 368 | 171 | 3191 |
> | GFNet-H-B | 482 | 367 | 144 | 2776 |
> | Sequencer2D-L (Ours) | 173 | 322 | 54 | 3516 |
>
> >  Are there any potential ways to reduce the computational cost, such as reducing the length of the sequence by downsampling? If so, can the authors report the model performance compared with the baselines under a similar scale of FLOPs and throughput?
>
> Please see Reply to Question 3. The accuracy of LSTM does not change when it is replaced by GRU; FLOPs and throughput improve slightly in that case (Section 4.3). Combining RNNs with local operations can indeed shorten the sequence length of input to the RNNs. It could improve throughput, but the demonstration is a future challenge.
>
> ### Reply to Question 2
>
> Objects and the visual patterns that comprise them are often distributed continuously in the image. Based on this observation, Sequnecer injects corresponding inductive bias by using vertical and horizontal LSTMs, which tend to guarantee continuous long-term dependencies. Such inductive bias can be represented by RNNs, but not by self-attentions. Token interactions extend beyond the straight line on which the LSTM acts: The interaction between any two tokens is formed by stacking two sequencing blocks. As for the impact of LSTM memory on the processing of spatial information, it is not straightforward to visualize the long-term dependence between tokens, unlike the case of attentions. We are convinced that this is a fascinating future work. Instead, our revised version visualizes each input-output tensor to BiLSTM2D for better understanding. From the hidden state visualization, it can be observed that the tokens processed in the vertical and horizontal directions interact to form two-dimensional spatial patterns. The closer tokens are in position, the stronger their interaction tends to be; the farther tokens are in position, the less their interaction tends to be (Figure 4).

---

> ### Author Response · Authors · 2022-08-02
> **Response to Reviewer GTX2 (2/2)**
>
> ### Reply to Question 3
>
> Thank you for your interest about the case of other RNNs. It is one of the questions that we have been wondering about too. We have reported the performances of models in which LSTM is replaced by other RNN-cells, including GRU, in Table 11b, P.19, Appendix in the submitted version. We have moved the result about replacing the RNN to the main paper.
>
> We reiterate the result:
>
> | Model           | \#Params. | FLOPs | Infer Throughput(image/s) | Acc. |
> | --------------- | --------- | ----- | ------------------------- | ---- |
> | RNN-Sequencer2D | 19M       | 5.8G  | 445                       | 80.6 |
> | GRU-Sequencer2D | 25M       | 7.5G  | 402                       | 82.3 |
> | Seqeucer2D-S    | 28M       | 8.4G  | 347                       | 82.3 |
>
> RNN-Sequencer2D replaces LSTM in Seqeucer2D-S with tanh-RNN, and GRU-Sequencer2D replaces LSTM in Seqeucer2D-S with GRU. The table suggests that all of these Metaformer-like architectures, including RNN-cell, are meaningful. Also, tanh-RNN performs slightly worse than others, probably due to its lower ability to model long-range dependence. LSTM does not outperform significantly in accuracy than GRU but do than tanh-RNN. Tanh-RNN is not entirely inaccurate: For example, it is better than the accuracy of RegNetY-4GF [1].
>
> #### Reference
>
> [1] "Designing network design spaces." CVPR 2020.

---

> > ### Comment · Reviewer_GTX2 · 2022-08-09
> > **Thanks for the response**
> >
> > Thanks for the response from the authors. My concerns are mostly addressed. Although I am still worried about the throughput issue in standard ImageNet resolution, I lean toward acceptance as the successful trial of replacing self-attention with LSTM in ViT deserves credit.

---

### Official Review · Reviewer_ouBL · 2022-07-11

**Rating:** 5
**Confidence:** 5
**Soundness:** 2 fair
**Presentation:** 3 good
**Contribution:** 2 fair

**Summary:**

This paper proposes an architecture for image classification named Sequencer, which utilizes the BiLSTM module to replace the self-attention module in the vision transformer model. The BiLSTM module is further improved by processing the vertical and horizontal axes in parallel from top/botton and left/right directions. Experiments on image classification tasks demonstrate that the proposed method can acheive similar performance with existing classification models with similar number of paramters.

**Questions:**

Please refer to the weaknesses part

**Limitations:**

The limitations are mainly about the limited novelty of the proposed method and the poor experimental results (much higher FLOPs, lack of experiments on other vision tasks).

**Strengths And Weaknesses:**

Strengths:

+ This paper is well-written. The idea is easy to understand.

+ The proposed method is the first work to empirically show the effectiveness of LSTM modules in large scale image classification tasks, which would have a board impact in investigating the potential of LSTM-like architectures in the computer vision field.

+ Ablations and visualization results are rich, which present the validity of the proposed method in terms of the importance of each component.

Weaknesses:

- The novelty of the proposed method is limited. The proposed Sequencer replaces the self-attention module in ViT with the existing BiLSTM module. Besides, [r1] shows that the self-attention module in ViT can be replaced with a simple spatial pooling operator, which suggests that such replacement is incremental.

- Although the proposed model can achieve similar performance with existing SOTA architecures, it requires much higher FLOPs and throughput as shown in Table 1.

- Evaluation is only conducted on image classification. It would be better to evaluate the proposed architecture on more vision tasks such as detection and segmetation to show its generalization ability.

[r1] MetaFormer Is Actually What You Need for Vision. CVPR 2022.

---

> ### Author Response · Authors · 2022-08-02
> **Response to Reviewer ouBL**
>
>
> You have raised important questions. We would like to thank you for this. Find below our answers to the reviewer's concerns:
>
> ### The novelty of the proposed method.
>
> As you say, in order to claim the novelty of our method, we need to state exactly how it differs from Metaformer[1]. We agree that Seuqnecer is a MetaFormer template-based work; however, we have adopted LSTM, unexplored non-local and non-attentional inductive bias, while MetaFormer(PoolFormer)[1] uses pooling which is local bias. In addition, LSTM is a different module from ViT[2], MLP-Mixer[3], and its many variants, which also follow the template of Metaformer[1]. This effort not only provides further evidence of the MetaFormer concept but also encourages the community to rethink the possibilities of LSTM-like architectures. Sequencer outperforms PoolFormer[1] in terms of performance other than throughput. Compared to PoolFormer-M36, Sequencer2D-S require the number of parameters to be about half and the throughput is about 70%, with 0.2 top-1 accuracy increases. It loses a bit in throughput but outperforms in top-1 accuracy and memory. From Table 1, compared to PoolFormer-M36, Sequencer2D-S require the number of parameters to be about half and the throughput is about 70%, with 0.2 top-1 accuracy increases. It loses a bit in throughput but outperforms in top-1 accuracy and memory. Thus, in terms of accuracy and number of parameter, Sequencer is superior to PoolFormer. This result also provides new evidence for the important hypothesis, MetaFormer.
>
>
> ### High FLOPs and low throughput
>
> As you point out, the drawback of this model is poor throughput, which is also mentioned in the paper as a limitation; improving throughput is certainly a subject for future research. The revised version more clearly states this.  As we currently know, the accuracy of LSTM does not change when it is replaced by GRU; FLOPs and throughput improve slightly in that case (Section 4.3, revised version):
>
> |Model|\#Params.|FLOPs|Infer Throughput(image/s)|Acc.|
> |---|---|---|---|---|
> |GRU-Sequencer2D|25M|7.5G|402|82.3|
> |Seqeucer2D-S|28M|8.4G|347|82.3|
>
> For further improvement, we need to consider shortening the sequence length of RNNs in combination with local operations such as pooling, or developing lightweight recurrent modules.
>
> ### The generalization ability in other tasks
>
> We thank the reviewer for their suggestion to add other visual tasks, including segmentation and detection experiments. The segmentation experiments have already been conducted in the rebuttal period. We have added the results presented below to Section 4.4 and Appendix C.4 in the revised version. We employed Seuqnecer as the backbone of SemanticFPN[4] to train and evaluate semantic segmentation. The dataset used is ADE20k[5], with a batch size of 32. AdamW[6] is used,  with the initial learning rate of 2e-4, the polynomial decay schedule with a power of 0.9, and 40000 training iterations. These settings follow the Metaformer settings[1]. The results are shown below:
>
> |Model|#Param.|mIoU|
> |---|---|---|
> |PVT-Small[7]|28.2|39.8|
> |PoolFormer-S24[1]|23.2|40.3|
> |Sequencer2D-S|31.6|46.1|
> |---|---|---|
> |PVT-Medium[7]|48.0|41.6|
> |PoolFormer-S36[1]|34.6|42.0|
> |Sequencer2D-M|42.3|47.3|
> |---|---|---|
> |PVT-Large[7]|65.1|42.1|
> |PoolFormer-M36[1]|59.8|42.4|
> |Sequencer2D-L|58.3|48.6|
>
> This result indicates that Sequencer's generalization ability for segmentation is comparable to other leading models. Studies of another tasks such as object detection are future research topics; We will be able to contain the detection experiment's results in the revised version by Aug. 10.
>
> #### Reference
>
> [1] "Metaformer is actually what you need for vision." CVPR 2022.
>
> [2] "An image is worth 16x16 words: Transformers for image recognition at scale." ICLR 2021.
>
> [3] "Mlp-mixer: An all-mlp architecture for vision." NeurIPS 2021.
>
> [4] "Panoptic Feature Pyramid Networks" CVPR 2019.
>
> [5] "Scene parsing through ade20k dataset." CVPR 2017.
>
> [6] "Decoupled weight decay regularization." ICLR 2019.
>
> [7] "Pyramid vision transformer: A versatile backbone for dense prediction without convolutions" ICCV 2021

---

### Official Review · Reviewer_gJFV · 2022-07-11

**Rating:** 6
**Confidence:** 4
**Soundness:** 3 good
**Presentation:** 3 good
**Contribution:** 3 good

**Summary:**

This paper introduces a new model architecture using LSTM for image classification. By adapting 2-dimensional LSTM (Bi-LSTM for vertical and horizontal directions) into the Transformer-like architecture, the model outperforms ViT-based and SOTA CNN-based architectures with less number of parameters.



**Questions:**

1. The authors mentioned that 'The higher the input resolution, the more memory-efficient and throughput-economical are on Sequencers' (Line 251-252 and Figure 4). I believe LSTM, in particular, 2D LSTM is not memory efficient and cannot be throughput-economical as it requires computing and saving the activation for all directions. Especially it should be worse for the higher resolutions. Could authors explain how sequencer becomes memory-efficient and throughput-economical?

2. It would be great to discuss and address the weaknesses mentioned above during the rebuttal.

**Limitations:**

The authors explained the limitations and potential negative societal impact of their work in the paper.

**Strengths And Weaknesses:**

## Strengths

1. The paper is clearly written.

2. The paper proposes a simple yet effective framework using LSTM. The model outperforms transformer and CNN-based models for image classification. This work provides a great alternative to Transformer and CNNs for image classification.

3. The proposed model is especially efficient for higher resolutions.

## Weaknesses

1. Lack of related work

- There are a number of studies using multi-directional LSTM/RNN for vision tasks that are very relevant to this work e.g., [1-4]. The authors should cite and discuss the similarities and differences.

- ReNet [69] is very relevant to this work. The authors pointed out that the major difference is to use a transformer-like block structure. However, the benefit of this structure and what it provides to the model compared to ReNet or other related works [1-4] are missing.

2. Due to LSTM's sequence nature, LSTM-based models are not easily parallelizable, especially compared to transformer and CNN-based models. I see that throughput is much worse than other models. I assume training time would be especially slow. It is unclear to me how throughput improves with higher resolutions.

[1] "Multi-dimensional recurrent neural networks." ICANN 2007.

[2] "Pixel recurrent neural networks." ICML 2016.

[3] "Scene labeling with lstm recurrent neural networks." CVPR 2015.

[4] "Semantic Object Parsing with Local-Global Long Short-Term Memory" CVPR 2016

---

> ### Author Response · Authors · 2022-08-02
> **Response to Reviewer gJFV (1/2)**
>
>
> Thank you for your comments and for pointing out further related work! We have included those papers and the discussion to resolve your suspicions in the paper. The revised paper has been uploaded to OpenReview.
>
> ### Reply to Question 1
>
> Thank you for your question. It might cause a misunderstanding to you due to our unclear statement. We have revised　the relevant part, as they should be claimed as memory economically and throughput-economically "compared to DeiT".
>
> > The higher the input resolution, the more memory-efficient and throughput-economical are on Sequencers
>
> In particular, the above is incorrect and is corrected below:
>
> > The higher the input resolution, the higher memory-efficiency and throughput of Sequencers when compared to DeiT.
>
> Why is Sequencers more memory economical than DeiT on high-resolution input? BiLSTM2D processes multiple columns and rows at once, using $WC/2$- and $HC/2$-dimensional memory cell state, respectively. BiLSTM2D hidden states are used as $CHW$-dimensional outputs. In contrast, a multi-head-attention requires $CHW$-dimensional value and $head*(HW)^2$-dimensional attention maps, where $H$, $W$, and $C$ is height, width, and channel, respectively. Thus, increasing H and W is disadvantageous to DeiT's memory consumption. In addition, Figure 3c(revised version) supports this view.
>
> At the $896^2$ resolution in Figure 3d(revised version), we see experimentally that the throughput of Sequencer is better than DeiT. This result is influenced by the vertical and horizontal decomposition, not the usual LSTM structure. Assuming $W=H$ for simplicity, the complexity of self-attention is $\mathcal{O}(W^4 C)$, whereas the computational complexity of BiLSTM is $\mathcal{O}(WC^2)$. Namely, the computational complexity of attention is $\mathcal{O}(W^3/C)$ times higher than that of BiLSTM. By contrast, there are $\mathcal{O}(1)$ sequential operations for self-attention, whereas there are $\mathcal{O}(W)$ sequential operations for BiLSTM2D. This implies that the increase in complexity of self-attention by increasing W has a much larger impact than the increase in BiLSTM2D sequence operations. Therefore, assuming we use a sufficiently efficient RNN cell implementation, such as official PyTorch LSTMs we are using, the increase of the complexity of self-attention is much more rapid than BiLSTM2D. It implies a lower throughput of self-attention compared to BiLSTM2D at high resolution.
>
> ### Reply to Question 2 (Lack of related work)
>
> Thank you for your suggestion to add the citations and the study's position compared to the paper.
>
> As you said, ReNet [5] is one of the excellent studies that are very relevant to our work. We have followed your suggestion and revised to include the following explanation.
>
> > ReNet [5] uses a 4-way LSTM and non-overlapping patches as input. In this respect, it is similar to Sequencer. Meanwhile, there are three differences. First, Sequencer is the first MetaFormer [6] realized by adopting LSTM as the token mixing block. Sequencer also adopts a larger patch size than ReNet [5]. The benefit of adopting these designs is that we can modernize LSTM-based vision architectures and fairly compare LSTM-based models with ViT. As a result, our results provide further evidence for the extremely interesting hypothesis MetaFormer. Second, connecting the vertical BiLSTM and the horizontal BiLSTM is different. Our work connects them in parallel, allowing us to gather vertical and horizontal information simultaneously, whereas ReNet [5] adopts the method of the output of the horizontal BiLSTM as input to the vertical BiLSTM). Finally, we trained Sequencer on large datasets such as ImageNet, whereas ReNet [5] is limited to small datasets as MNIST, CIFAR-10, and SVHN, and has not shown the effectiveness of LSTM for larger datasets.
>
> For [1-4], which you have pointed out, the revised version compares these works with our study. The submitted version already has cited [1] but has been revised to show more differences in the revised version.

---

> ### Author Response · Authors · 2022-08-02
> **Response to Reviewer gJFV (2/2)**
>
> ### Reply to Question 2 (About Throughput)
>
> Throughput itself is generally considered to decrease as resolution increases, but when throughput is compared to DeiT, Sequencer has an advantage as resolution increases. We have corrected the incomplete description in the revised version. The throughput advantage of Sequencer over DeiT, as the resolution increases is as mentioned in the reply to Question 1.
>
> Your concern about the time it takes to train is most understandable. The following are the peak memory results during training.
>
> | Model | Infer Throughput(image/s) | Infer Peak Mem. | Train Throughput(image/s) | Train Peak Mem. |
> | --- | --- | --- | --- | --- |
> | RegNetY-4GF | 823 | 225 | 228 | 1136 |
> | ConvNeXt-T | 1124 | 248 | 337 | 1418 |
> | DeiT-S | 1569 | 180 | 480 | 1195 |
> | Swin-T | 894 | 308 | 268 | 1613 |
> | ViP-S/7 | 702 | 195 | 214 | 1587 |
> | CycleMLP-B2 | 586 | 234 | 158 | 1357 |
> | PoolFormer-S24 | 988 | 183 | 313 | 1461 |
> | Sequencer2D-S (Ours) | 347 | 196 | 110 | 1799 |
> | RegNetY-8GF | 751 | 333 | 211 | 1776 |
> | T2T-ViT$_{t}$-19 | 654 | 1140 | 197 | 3520 |
> | CycleMLP-B3 | 367 | 287 | 100 | 2326 |
> | PoolFormer-S36 | 673 | 220 | 213 | 2187 |
> | GFNet-H-S | 755 | 282 | 227 | 1740 |
> | Sequencer2D-M (Ours) | 270 | 244 | 83 | 2311 |
> | RegNetY-12GF | 695 | 440 | 199 | 2181 |
> | ConvNeXt-S | 717 | 341 | 212 | 2265 |
> | Swin-S | 566 | 390 | 165 | 2635 |
> | Mixer-B/16 | 1011 | 407 | 338 | 1864 |
> | ViP-M/7 | 395 | 396 | 130 | 3095 |
> | CycleMLP-B4 | 259 | 338 | 70 | 3272 |
> | PoolFormer-M36 | 496 | 368 | 171 | 3191 |
> | GFNet-H-B | 482 | 367 | 144 | 2776 |
> | Sequencer2D-L (Ours) | 173 | 322 | 54 | 3516 |
>
> The results above were measured under the same conditions as the throughput measurements in Table 1. While it is true that training throughput is not good, the results show that the training throughput is about three times the inference throughput for all these models. Compared to other models, both measured inference and training time are not good. Future research should be performed to determine if throughput can be improved by reducing sequence length in combination with convolution and pooling.
>
> #### Reference
>
> [1] "Multi-dimensional recurrent neural networks." ICANN 2007.
>
> [2] "Pixel recurrent neural networks." ICML 2016.
>
> [3] "Scene labeling with lstm recurrent neural networks." CVPR 2015.
>
> [4] "Semantic Object Parsing with Local-Global Long Short-Term Memory" CVPR 2016.
>
> [5] "Renet: A recurrent neural network based alternative to convolutional networks." arXiv:1505.00393 2015.
>
> [6] "Metaformer is actually what you need for vision." CVPR 2022.

---

> > ### Comment · Reviewer_gJFV · 2022-08-09
> > **response**
> >
> > Thank you for the answers and updating the paper.
> >
> > Please add the detailed discussion about training/inference throughput and memory in the final version as this is a major limitation of the proposed method. Also, I suggest the authors make the code and pre-trained models public. This will help the community to reconsider the RNN-based models for vision tasks.
> >
> > I don't have any further questions.

---

> > > ### Author Response · Authors · 2022-08-09
> > > **Thank you for the response.**
> > >
> > > Thanks for your positive comments. As you suggested, we will include the training/inference throughput discussion and the official code link in the final version.
> > >
> > > Thanks to your excellent review, we were able to improve the manuscript.

---

### Author Response · Authors · 2022-08-02
**To all concerned**

We thank the reviewers for their insightful comments on our paper. The comments have helped us to improve the paper significantly. As for the values of Sequencer2D-L throughput and memory consumption during inference in Table 1, we inadvertently wrote worse values than the actual values, and we have corrected those values. The revised paper will be uploaded to OpenReview.

---

### Meta-Review · Area_Chair_gwFf · 2022-08-24

**Recommendation:** Accept
**Confidence:** Less certain

**Metareview:**

Four reviewers provided detailed feedback on this paper. The authors responded to the reviews and I appreciate the authors' comments and clarifications, specifically that each question/comment is addressed in detail. The authors also uploaded a revised version of the paper.

After the two discussion periods, all four reviewers suggest to accept the paper (although the scores do not exceed a "weak accept"). After considering the reviewers' and authors' comments, I believe that the paper should be accepted to NeurIPS.

Weaknesses include:
* Some concerns about experimental results, e.g. highlighting accuracy vs. number of parameters but not also highlighting limitations when looking throughput (comparing only parameters (or FLOPS) can sometimes be misleading, see also [The efficiency misnomer, ICLR22](https://arxiv.org/abs/2110.12894)). But it's good that throughput numbers are presented in the paper and the paper acknowledges this limitation. Related: concerns about computational cost.
* Some concerns regarding relevant related literature (addressed in comments and revision) and novelty of the approach.
* Limitation to image classification only in the experiments (partially addressed in comments and revision).
* More interpretation of the effect of using LSTMs could be helpful to the reader (partially addressed in comments).

Strengths include:
* Interesting, conceptually simple approach that revisits LSTMs for images, which could be specifically useful for high resolution images.
* Reviewers agree that the paper is well-written.
* Experimental results and ablations are strong with respect to the claims made.

Minor points (not affecting this decision, but potentially useful to authors when preparing the final revision):
* MLP-based methods "cannot cope with flexible input sizes during inference" - I think this is only partially true, even the original MLP-Mixer paper shows how this can be solved e.g. in fine-tuning by "modifying the shape of Mixer’s token-mixing MLP blocks"
* minor typo I randomly encountered: Table 3, row 3, column "Flowers" 89.5 -> 98.5
* "It is demonstrated that modeling long-range dependencies by self-attention is not necessarily essential in computer vision" - To some degree similar "demonstrations" are visible in CNNs and MLP-Mixers, so this claim seems a bit strong, maybe?

**Award:**

No

---

### Decision · Program_Chairs · 2022-09-14

Accept